# BEHAVIOR: Benchmark for Everyday Household Activities in Virtual, Interactive, and Ecological Environments

**Sanjana Srivastava\*♠, Chengshu Li\*♠, Michael Lingelbach\*♡ Roberto Martín-Martín\*♠**
**Fei Xia♣, Kent Vainio♠, Zheng Lian♠, Cem Gokmen♠, Shyamal Buch♠**
**C. Karen Liu♠★, Silvio Savarese♠★, Hyowon Gweon◇★, Jiajun Wu♠★, Li Fei-Fei♠★**

Department of Computer Science♠, Neurosciences IDP♡, Electrical Engineering♣, Psychology◇
Institute for Human-Centered AI (HAI)★
Stanford University

**Abstract:** We introduce BEHAVIOR, a benchmark for embodied AI with 100 activities in simulation, spanning a range of everyday household chores such as cleaning, maintenance, and food preparation. These activities are designed to be realistic, diverse and complex, aiming to reproduce the challenges that agents must face in the real world. Building such a benchmark poses three fundamental difficulties for each activity: definition (it can differ by time, place, or person), instantiation in a simulator, and evaluation. BEHAVIOR addresses these with three innovations. First, we propose a predicate logic-based description language for expressing an activity's initial and goal conditions, enabling generation of diverse instances for any activity. Second, we identify the simulator-agnostic features required by an underlying environment to support BEHAVIOR, and demonstrate in one such simulator. Third, we introduce a set of metrics to measure task progress and efficiency, absolute and relative to human demonstrators. We include 500 human demonstrations in virtual reality (VR) to serve as the human ground truth. Our experiments demonstrate that even state-of-the-art embodied AI solutions struggle with the level of realism, diversity, and complexity imposed by the activities in our benchmark. We make BEHAVIOR publicly available at behavior.stanford.edu to facilitate and calibrate the development of new embodied AI solutions.

**Keywords:** Embodied AI, Benchmarking, Household Activities

## 1 Introduction

Embodied AI refers to the study and development of artificial agents that can perceive, reason, and interact with the environment with the capabilities and limitations of a physical body. Recently, significant progress has been made in developing solutions to embodied AI problems such as (visual) navigation [1–5], interactive Q&A [6–10], instruction following [11–15], and manipulation [16–22].

To calibrate the progress, several lines of pioneering efforts have been made towards benchmarking embodied AI in simulated environments, including Rearrangement [23, 24], TDW Transport Challenge [25], VirtualHome [26], ALFRED [11], Interactive Gibson Benchmark [27], MetaWorld [28], and RLBench [29], among others [30–32]. These efforts are inspiring, but their activities represent only a fraction of challenges that humans face in their daily lives. To develop artificial agents that can eventually perform and assist with everyday activities with human-level robustness and flexibility, we need a comprehensive benchmark with activities that are more **realistic**, **diverse**, and **complex**.

But this is easier said than done. There are three major challenges that have prevented existing benchmarks from accommodating more realistic, diverse, and complex activities:

- Definition: Identifying and defining meaningful activities for benchmarking;
- Realization: Developing simulated environments that realistically support such activities;

---

*indicates equal contribution
correspondence to {sanjana2,chengshu,mjlbach,robertom}@stanford.edu

5th Conference on Robot Learning (CoRL 2021), London, UK.

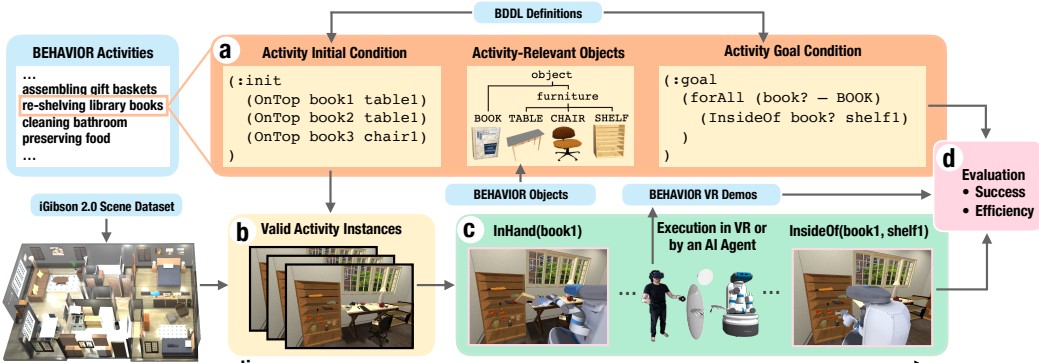

Figure 1: **Benchmarking Embodied AI with BEHAVIOR**: ⓐ We define 100 realistic household activities from the American Time Use Survey [34] and define them with a set of relevant objects, organized with WordNet [35], and logic-symbolic initial and goal conditions in BDDL (Sec. 4). ⓑ We provide an implementation of BEHAVIOR in iGibson 2.0 that generates potentially infinite diverse activity instances in realistic home scenes using the definition. ⓒ AI agents perform the activities in simulation through continuous physical interactions of an embodied avatar with the environment. Humans can perform the same activities in VR. BEHAVIOR includes a dataset of 500 successful VR demonstrations. ⓓ Changes in the scene are continuously mapped to their logic-symbolic equivalent representation in BDDL and checked against the goal condition; we provide intermediate success scores, metrics on agent's efficiency, and a human-centric metric relative to the demonstrations.

- Evaluation: Defining success and objective metrics for evaluating performance.

We propose **BEHAVIOR** (Fig. 1)–**B**enchmark for **E**veryday **H**ousehold **A**ctivities in **V**irtual, **I**nteractive, and ec**O**logical envi**R**onments, addressing the three key challenges with three technical innovations. First, we introduce BEHAVIOR Domain Definition Language (BDDL), a representation adapted from predicate logic that maps simulated states to semantic symbols. It allows us to define 100 activities as initial and goal conditions, and enables generation of potentially infinite initial states and solutions for achieving the goal states. Second, we facilitate its realization by listing environment-agnostic functional requirements for realistic simulation. With proper engineering, BEHAVIOR can be implemented in many existing environments; we discuss a fully functional instantiation in iGibson 2.0 [33] in this paper including the necessary object models (1217 models of 391 categories). Third, we provide a comprehensive set of metrics to evaluate agent performance in terms of success and efficiency. To make evaluation comparable across diverse activities, scenes, and instances, we propose a set of metrics relative to demonstrated human performance on each activity, and provide a large-scale dataset of 500 human demonstrations (1077.7 min) in virtual reality, which serve as ground truth for evaluation and may also facilitate developing imitation learning solutions.

BEHAVIOR's 100 activities are realistic, diverse, and complex. They are often performed by humans in their homes (e.g., cleaning, packing or preparing food) and require long-horizon solutions for changing not only the position of multiple objects but also their internal states or texture (e.g., temperature, wetness or cleanliness levels). As we demonstrate by experimentally evaluating the performance of two state-of-the-art reinforcement learning algorithms (Section 7), these properties make BEHAVIOR activities extremely challenging for existing solutions. By presenting well-defined challenges beyond the capabilities of current solutions, BEHAVIOR can serve as a unifying benchmark that guides the development of embodied AI.

## 2 Related Work

Benchmarks and datasets have played a critical role in recent impressive advances in AI, particularly computer vision. Image [36–39] and video datasets [40–45] enable study and development of solutions for important research questions by providing both training data and fair comparison. These datasets, however, are passive observations that are not well-suited for development of embodied AI that must control and understand the consequences of their own actions.

**Benchmarks for Embodied AI:** Although real-world challenges [46–53] provide the ultimate testbed for embodied AI agents, benchmarks in simulated environments serve as useful alternatives with several advantages; simulation enables faster, safer learning, and supports more reproducible,

**Table 1 — Comparison of Embodied AI Benchmarks**

| | BEHAVIOR | AI2THOR Vis. Room Rearr. | TDW Transport | Rearrangement T5 (Habitat) | ManipulaTHOR ArmPointNav | Interactive Gibson Benchmark | VirtualHome | ALFRED | Rearrangement T2 (OCRTOC) | IKEA Furniture Assembly | RLBench | Metaworld | Robosuite | SoftGym | DeepMind Control Suite | OpenAIGym | Habitat 1.0 | Gibson |
|---|---|---|---|---|---|---|---|---|---|---|---|---|---|---|---|---|---|---|
| | | | Mobile manipulation | | | | | | | | Static manipulation | | | | | | Navigation | |
| **Realism** — Activity selections reflect human behavior | ✓ | ✗ | ✗ | ✗ | ✗ | ✗ | ✗ | ✗ | ✗ | ✗ | ✗ | ✗ | ✗ | ✗ | ✗ | ✗ | ✗ | ✗ |
| Kinematics, dynamics | ✓ | ✓ | ✓ | ✓ | ✓ | ✓ | ✗ | ✓ | ✓ | ✓ | ✓ | ✓ | ✓ | ✓ | ✓ | ✓ | ✓ | ✓ |
| Continuous extended states (e.g. temp., wetness) | ✓ | ✗ | ✗ | ✗ | ✗ | ✗ | ✗ | ✗ | ✗ | ✗ | ✗ | ✗ | ✗ | ✗ | ✗ | ✗ | ✗ | ✗ |
| Changing flexible materials | ✗ | ✗ | ✗ | ✗ | ✗ | ✗ | ✗ | ✗ | ✗ | ✗ | ✗ | ✗ | ✗ | ✗ | ✗ | ✗ | ✗ | ✗ |
| Realistic action execution | ✓ | ✗ | ✓ | ✗ | ✓ | ✓ | ✗ | ✗ | ✓ | ✓ | ✓ | ✓ | ✓ | ✓ | ✓ | ✓ | ✓ | ✓ |
| Scenes reconstructed from real homes | ✓ | ✗ | ✗ | ✓ | ✗ | ✓ | ✗ | ✗ | ✗ | ✗ | ✗ | ✗ | ✗ | ✗ | ✗ | ✗ | ✓ | ✓ |
| **Diversity** — # Activities | 100 | 1 | 1 | 1 | 1 | 2 | 549 | 7 | 5 | 100 | 50 | 1 | 5 | 10 | 28 | 8 | 2 | 3 |
| Infinite scene-agnostic instantiation | ✓ | ✗ | ✗ | ✗ | ✗ | ✗ | ✗ | ✗ | ✗ | ✗ | ✗ | ✗ | ✗ | ✗ | ✗ | ✗ | ✗ | N/A |
| Object categories | 391 | 118 | 50 | YCB | 150 | 5 | 509 | 84 | 12 | 73+ | 28 | 7 | 10 | 4 | 4 | 4 | Matterport | N/A |
| Object models | 1217 | 118 | 112 | YCB | 150 | 152 | - | 84 | 101 + YCB | 73+ | 28 | 80 | 10 | 4 | 4 | 4 | N/A | N/A |
| Scenes / Rooms | 15 / 100 | - / 120 | 15 / 90-120 | 55 static / - | - / 30 | 10 / - | 7 / - | - / 120 | 1 / - | 1 / - | 1 / - | 1 / - | 1 / - | 1 / - | 1 / - | 1 / - | Matterport + Gibson | 572 static |
| **Complexity** — Activity length²(steps) | 300-20000 | <100 | 100-1000 | 100-1000 | <100 | 100-1000 | <100 | <100 | 100-1000 | <1000 | <100 | <100 | <100 | <100 | <100 | <100 | <100 | 100-1000 |
| Objs. per activity | 3-34 | 5 | 7-9 | 2-5 | 2-3 | 10 | 1-24 | 2 | 5-10 | 1-2 | 1-2 | 1 | 1-3 | 1-3 | 1-3 | 1 | 0-1 | N/A |
| Benchmark focus: Task-Planning and/or Control | TP+C | TP | TP+C | TP+C | TP+C | C | TP | TP | TP+C | C | TP+C | C | C | C | C | C | C | C |
| Diff. state changes required per activity (see A.2) | 2-8 | 4 | 4 | 4 | 2 | 1-3 | 1-7 | 2-3 | 1 | 1-3 | 1-4 | 4 | 1 | 1-3 | 1-2 | 1-2 | 1 | 1 |
| # Human VR demos | 500 | 0 | 0 | 0 | 0 | 0 | 0 | 0 | 0 | 0 | 0 | 0 | 0 | 0 | 0 | 0 | 0 | 0 |

[1]Estimate of a near-optimal, e.g. human, execution of the activity given the platform's action space

Table 1: **Comparison of Embodied AI Benchmarks:** BEHAVIOR activities are exceptionally realistic due to their grounding in human population time use [34] and realistic simulation (sensing, actuation, changes in environment) in iGibson 2.0. The activity set is diverse in topic, objects used, scenes done in, and state changes required. The diversity is reinforced by the ability to generate infinite new instances scene-agnostically. BEHAVIOR activities are complex enough to reflect real-world housework: many decision steps and objects in each activity. This makes BEHAVIOR uniquely well-suited to benchmark task-planning and control, and it is the only one to include human VR demonstrations (see Table A.1 for more detail).

accessible, and fair evaluation. However, in order to serve as a meaningful proxy for real-world performance, simulation benchmarks need to achieve high levels of 1) **realism** (in the activities, the models, the sensing and actuation of the agent), 2) **diversity** (of scenes, objects and activities benchmarked), and 3) **complexity** (length, number of objects, required skills and state changes). Below we review existing benchmarks based on these three criteria (see Table 1 for a summary).

Benchmarks for *visual navigation* [54, 55] provide high levels of visual realism and diversity of scenes, but they often lack interactivity or diversity of activities. The Interactive Gibson Benchmark [27] trades off some visual realism for physically realistic object manipulation in order to benchmark interactive visual navigation. While benchmarks for *stationary manipulation* [56, 29, 28, 30, 57, 31, 32] fare well on physical realism, they commonly fall short on diversity (of scenes, objects, tasks) and complexity (often having simple activities that take a few seconds). Benchmarks for *instruction following* [11, 26] provide diversity of scenes, objects and possible changes of the environment, but with low levels of complexity; the horizon of the activities is shorter as the agents decide among a discrete set of predefined action primitives with full access to the state of the world.

Closer to BEHAVIOR, a recent group of benchmarks has focused on *rearrangement tasks* [23–25] in realistic simulation environments with diverse scenes. The initial Rearrangement position paper [23] poses critical questions such as how to define embodied AI tasks and measure solution quality. Importantly however, most household activities go far beyond the scope of rearrangement (see comparison in Fig. A.2). While such focus can inspire new solutions for solving rearrangement tasks, these solutions may not generalize to activities that require more than physical manipulation of object coordinates. Indeed, the majority of household activities involve other state changes (cooking, washing, etc.) (Fig. A.2, [34]). BEHAVIOR therefore incorporates 100 activities that humans actually spend time on at home [34] (Sec. 3). To express such diverse activites in a common language, we present a novel logic-symbolic representation that defines activities as initial and goal states, inspired by but distinct from the Planning Domain Definition Language (PDDL) [58] (see Sec. 4). These definitions yield in principle infinite instances per activity and accept any meaningful solution. We implement activity-independent metrics including a human-centric metric normalized to human performance; to facilitate comparison and development of new solutions, we also present a dataset of 500 successful VR demonstrations.

# 3 BEHAVIOR: Benchmarking Realistic, Diverse, Complex Activities

Building on the advances led by existing benchmarks, BEHAVIOR aims to reach new levels of realism, diversity, and complexity by using household activities as a domain for benchmarking AI. See Table 1 for comparisons between BEHAVIOR and existing benchmarks.

**Realism in BEHAVIOR Activities:** To effectively benchmark embodied AI agents in simulation, we need realistic activities that pose similar challenges to those in the real world. BEHAVIOR achieves this by using a data-driven approach to identify activities that approximate the true distribution of real household activities. To this end, we use the American Time Use Survey (ATUS, [34]): A survey from the U.S. Bureau of Labor Statistics on how Americans spend their time. BEHAVIOR activities come from, and are distributed similarly to, the full space of simulatable activities in ATUS (see Fig. A.2). The use of an independently curated source of real-world activities is a unique strength of BEHAVIOR as a benchmark that reflects natural behaviors of a large population.

BEHAVIOR also achieves realism by simulating these activities in reconstructions of real-world homes. We use iGibson 2.0, a simulation environment with realistic physics simulation from the Bullet [59] physics engine and high-quality virtual sensor signals (see Fig. A.7), which includes 15 ecological, fully interactive 3D models of real-world homes with furniture layouts that approximate their real counterparts. These scenes are further populated with object models created by professional artists from the new BEHAVIOR Object dataset, which includes 1217 models of 391 categories grounded in the WordNet [35] taxonomy. The dataset covers a data-driven selection of activity-related objects (see Fig. A.8). Figs. A.10 and A.9 illustrate examples of objects and taxonomic arrangement. The 100 BEHAVIOR activities, visualized in Fig. A.1, go beyond comparable benchmarks that evaluate a few hand-picked activities in less realistic setups (see Table 1 Realism). iGibson 2.0 also provides a wide variety of realistic simulated robots that have real-world counterparts, e.g. LoCoBot, Quadrotor, Fetch, the last of which we can use to fulfill the BEHAVIOR activities (see Sec. 5).

**Diversity in BEHAVIOR Activities:** Benchmarks with diverse activities demand generalizable solutions. In real-world homes, agents encounter a range of activities that differ in 1) the capabilities required for achieving them, 2) the environments in which they occur (e.g., scenes, objects), and 3) the initial states of a particular scene. BEHAVIOR presents extensive diversity in all these dimensions. We include 100 activities that require a wide variety of state changes (e.g., moving objects, soaking materials, cleaning surfaces, heating/freezing food) demanding a broad set of agent capabilities (see Fig A.2). To reflect the diversity in the ways humans encounter, understand, and accomplish these activities, we provide two example definitions per activity.

BDDL, our novel representation for activity definition, allows new valid instances to be sampled from each definition, providing potentially infinite number of instances per activity. The resulting instances vary over scene, object models, and configuration, supported by implementation in iGibson 2.0 and BEHAVIOR Object dataset. Related benchmarks focus on fewer tasks, mostly limited to kinematic state changes and with scene- or position-constant instantiation (see Table 1 Diversity).

**Complexity in BEHAVIOR Activities:** Beyond diversity across activities, BEHAVIOR also raises the complexity of the activities themselves by benchmarking full household activities that parallel the length (number of steps an agent needs), the number of objects involved, and the number of required capabilities of real-world chores (see Fig. A.3, comparison in Table 1 Complexity). Compared to activities in existing benchmarks, these activities are very long-horizon with some requiring several thousand steps (even for humans in VR; see Fig. A.12), involve more objects (avg. 10.5), and require a heterogeneous set of capabilities (range: 2 - 8) to change various environment states.

## 4 Defining Realistic, Diverse, and Complex Household Activities with BDDL

BEHAVIOR challenges embodied AI agents to achieve a diverse set of complex long-horizon household activities through physical interactions in a realistically simulated home environment.

Adopting the common formalism of partially-observable Markov decision processes (POMDP), each activity has a state space $\mathcal{S}$ (see more details in A.3.2).

We define an *activity* $\tau$ as two sets of states, $\tau = \{S_{\tau,0}, S_{\tau,g}\}$, where $S_{\tau,0}$ is a set of possible initial states and $S_{\tau,g}$ is a set of acceptable goal states. In an *activity instance*, the agent must change the world state from some concrete $s_0 \in S_{\tau,0}$ to any $s_g \in S_{\tau,g}$. However, describing activities in the physical state space generates scene- or pose-specific definitions (e.g., [23, 30, 29]) that are far more specific than how humans represent these activities, limiting the diversity and complexity of existing embodied AI benchmarks. To overcome this, we introduce *BEHAVIOR Domain Definition Language* (BDDL), a predicate logic-based language that establishes a symbolic state representation built on

predefined, meaningful predicates grounded in simulated physical states; its variables and constants represent object categories from the BEHAVIOR object dataset. Each activity is defined in BDDL as an initial and goal condition parametrizing sets of possible initial states and satisfactory goal states $\bar{S}_{\tau,0}$ and $\bar{S}_{\tau,g}$. BDDL predicates create symbolic counterparts of the physical state, $\bar{S}$ (see Fig. 2).

BDDL overcomes limitations that hinder diversity through two mechanisms: first, an initial condition maps to infinite physical states in diverse scenes. Second, a goal condition detects all semantically satisfactory solutions. By contrast, other benchmarks support either infinite distinct instantiations but only in one scene per definition, because they sample from hard-coded regions; or instantiation in multiple scenes, but not infinitely because object poses are hard-coded on furniture objects in those scenes. BEHAVIOR is the only benchmark with both. BEHAVIOR also includes a systematic generation pipeline (Sec. A.3.3) enabling unlimited definitions per activity to

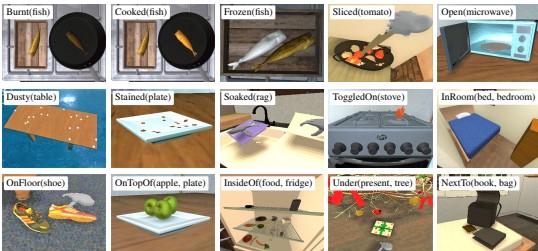

Figure 2: **Unary and Binary Predicates in BDDL:** We represent object states and relationships to other objects based on their kinematics, temperature, wetness level and other physical and functional properties, enabling a diverse and complex set of realistic activities

formalize the subjectivity of household activities. We include 200 definitions and 100 instances in simulation (Sec. 5). BEHAVIOR is thus the only benchmark equipped to formalize unlimited human-defined versions of an activity and create practically infinite unique instantiations in any scene. Finally, BEHAVIOR has purely declarative definitions of initial and goal condition, whereas some benchmarks provide imperative plans for getting from initial to goal [26]. The declarative nature creates a true test of an agent's capability of task planning.

## 5 Instantiating BEHAVIOR in a Realistic Physics Simulator

While BEHAVIOR is not bounded to any specific simulation environment, there are a set of functional requirements to be able to simulate BEHAVIOR activities: 1) maintain an object-centric representation (object identities enriched with properties and states), 2) simulate physical forces and motion, and generate virtual sensor signals (images), 3) simulate additional properties per object (e.g. temperature, soak level, cleanliness level) necessary for BEHAVIOR activities, 4) implement functionality to **generate** valid instances based on the literals defining an activity's initial condition, e.g., instantiating an object `insideOf` another, and 5) implement functionality to **evaluate** the atomic formulae relevant to the goal condition, e.g. checking whether an object is `cooked` or `onTopOf` another.

While BEHAVIOR activities are not tailored to a specific embodiment, we propose two concrete bodies to fulfill the activities offering different action spaces (see Fig. 1): a *bimanual humanoid* avatar (24 degrees of freedom, DoF), and a *Fetch robot* (12/13 DoF), both capable of navigating, grasping and interacting with the hand(s). Humans in VR embody the bimanual humanoid.

Because it models a real-world robot, agents trained with the Fetch embodiment could be directly tested with a real-world version of the hardware (see discussion on sim2real in Sec. A.8). Both agents receive sensor signals from the on-board virtual sensors, and perform actions at 30 Hz.

We provide a fully functional implementation of BEHAVIOR using iGibson 2.0, a new version of the open-source simulation environment iGibson that fulfills the requirements above. iGibson 2.0 provides an object-centric representation with additional properties, support for sources of heat and water, dust and stain particles, and changes in object appearance based on extended states. We also implement the two embodiments mentioned above and a set of action primitives inspired by [25, 55, 60, 24] to facilitate solution prototyping and task-planning research. The primitives are executing sequences of low-level actions resulting from a motion planning process (bilateral RRT* [61]) to `navigateTo`, `grasp`, `placeOnTop`, `placeInside`, `open`, and `close` the object given as argument. Further details can be found in Sec. A.4 and in [33]. Our implementation of BEHAVIOR in iGibson 2.0 goes beyond the capabilities of existing benchmarks and amplifies realism, diversity, and complexity (see Table 1).

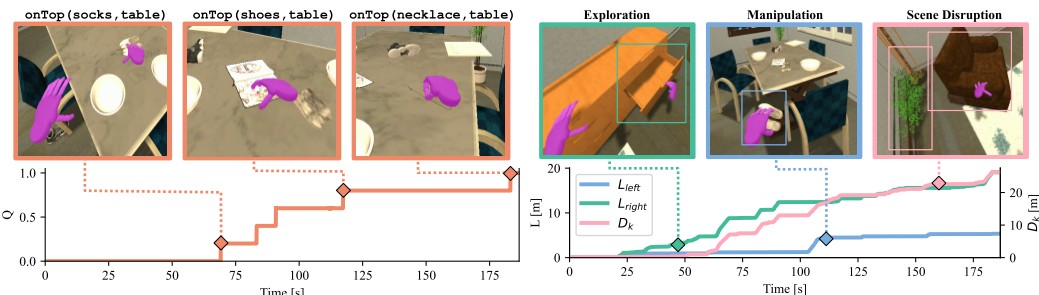

Figure 3: **Evaluation of human performance in `collect_misplaced_items`:** (*Left*) success score, $Q$; (*Right*) efficiency metrics: kinematic disarrangement, ($D_k$, dotted), hand interaction displacement ($L_{right}$, green, and $L_{left}$, blue); frames at the top depict significant events detected by the metrics; the success score detects the completion of activity-relevant steps; exploration, manipulation and scene disruption events are captured by the efficiency metrics that provide complementary information about the performance of the agent.

## 6 Evaluation Metrics: Success, Efficiency and Human-Centric Metric

BEHAVIOR provides evaluation metrics to quantify the performance of an embodied AI solution. Extending prior metrics suggested for Rearrangement [23], we propose a primary metric based on success and several secondary metrics for characterizing efficiency.

**Primary Metric – Success Score Q:**    The main goal of an embodied AI agent in BEHAVIOR is to perform an activity successfully (i.e., all logical expressions in the goal condition are met). A binary definition of success, however, only signals the end of a successful execution and cannot assess interim progress. To provide more guidance to agents and enable comparisons of partial solutions, we propose **success score** as the primary metric, defined as the **maximum fraction of satisfied goal literals in a ground solution to the goal condition** at each step. More formally:

Given an activity $\tau$ with goal state set $\bar{S}_{\tau,g}$, its goal condition can be flattened to a set $C$ of conjunctions $C_i$ of ground literals $l_{j_i}$.

For any $C_i \in C$, if all $l_{j_i} \in C_i$ are true then the goal condition is satisfied (see A.3.2 for definitions and technical details on flattening), i.e. for some current environment state $s$, we have $\bigvee_{C_i} \bigwedge_{l_{j_i}} l_{j_i} =$ True $\implies s \in \bar{S}_{\tau,g}$ . We compute the fraction of literals $l_{j_i}$ that are True for each $C_i$, and define the overall success score by taking the maximum: $Q = \max_C \frac{|\{l_{j_i} | l_{j_i} = \text{True}\}|}{|C_i|}$, where $|\cdot|$ is set cardinality.

An activity is complete when all literals in *at least one* $C_i$ of its goal condition are satisfied, achieving $Q = 1$ (100%). Fig. 3, left, depicts time evolution of $Q$ during an activity execution. $Q$ extends the fraction of objects in acceptable poses proposed as metric in [23], generalized to any type of activity.

**Secondary Metrics – Efficiency:**    Beyond success, efficiency is critical to evaluation; a successful solution in real-world tasks may be ineffective if it takes too long or causes scene disruption. We propose six secondary metrics that complement the primary metric (see Fig. 3, right, for examples):

- *Simulated time*, $T_{sim}$: Accumulated time in simulation during execution as the number of simulated steps times the average simulated time per step. $T_{sim}$ is independent of the computer used.
- *Kinematic disarrangement*, $D_K$: Displacement caused by the agent in the environment. This can be *accumulated* over time, or *differential*, i.e. computed between two time steps, e.g. initial, final.
- *Logical disarrangement*, $D_L$: Amount of changes caused by the agent in the logical state of the environment. This can be *accumulated* over time or *differential* between two time steps.
- *Distance navigated*, $L_{body}$: Accumulated distance traveled by the agent's base body. This metric evaluates the efficiency of the agent in navigating the environment.
- *Displacement of hands*, $L_{left}$ and $L_{right}$: Accumulated displacement of each of the agent's hands while in contact with another object for manipulation (i.e., grasping, pushing, etc). This metric evaluates the efficiency of the agent in its interaction with the environment.

These efficiency metrics above can be quantified in absolute units (e.g., distance, time) for scene- and activity-specific comparisons (**general efficiency**). To enable fair comparisons cross diverse activities in BEHAVIOR, we also propose normalization relative to human performance (**human-centric**

**efficiency**); given a human demonstration for an activity instance in VR, each secondary metric can be expressed as a *fraction of the maximum human performance* on that metric.

For this purpose, we present the BEHAVIOR Dataset of Human Demonstrations with 500 successful demonstrations of BEHAVIOR activities in VR (1077.7 min). Humans are immersed in iGibson 2.0, controlling the same embodiment used by the AI agents (details in Sec. A.6). The dataset includes a complete record of human actions including manipulation, navigation, and gaze tracking data (Fig. A.12, Fig. A.14, and Fig. A.16), supporting analysis and subactivity segmentation (Fig. A.11). Sec. A.6.2 presents a comprehensive analysis of these data; we quantify human performance in BEHAVIOR efficiency metrics (see Fig. A.12), and Fig. A.13 provides a further decomposition of room occupancy and hand usage across each BEHAVIOR activity. To our knowledge, this is the largest available dataset of human behavior in VR; these data can facilitate development of new solutions for embodied AI (e.g., imitation learning) and also support studies of human cognition, planning, and motor control in ecological environments.

# 7 Evaluating Reinforcement Learning in BEHAVIOR

In this section, we aim to experimentally demonstrate the challenges imposed by BEHAVIOR's realism, diversity, and complexity on state-of-the-art embodied AI solutions. BEHAVIOR is a benchmark for all kinds of embodied AI methods. Here, we evaluate two reinforcement learning (RL) algorithms that have excelled on simpler embodied AI tasks [62, 63, 21, 64–68]: Soft-Actor Critic (SAC [16]) and Proximal-Policy Optimization (PPO [17]). We use SAC to train policies in the original low-level continuous action space of the agent, and PPO for experiments using our implemented action primitives (for details on the agents, see Sec. 5). Due to limited computational resources, we evaluate on the 12 most simple activities (by distinct state changes involved) using the bimanual humanoid embodiment.

Reward is given by our staggered success score $Q$. We use as input a subset of the realistic agent's observations, RGB, depth and proprioception (excluding LiDAR, segmentation, etc.). Sec. A.7 includes more experimental details.

**Results in the original activities:** The first row of Table 2 shows the results of SAC (mean $Q$ at the end of training for 3 seeds) on the original 12 activities with the standard setup: realistic robot actions and onboard sensing. Even for these "simpler" activities, BEHAVIOR is too great a challenge: the training agents do not fulfill any predicate in the goal condition ($Q = 0$). In the following, we will analyze how each dimension of difficulty (realism, diversity, complexity) contributes to these results.

**Effect of complexity (activity length):** First, we evaluate the impact of the activity complexity (time length) on performance. We begin with an RL algorithm using our implemented action primitives based on motion planning. These temporally extended actions effectively shorten the horizon and length of the activity. The results of training with PPO are depicted in the second row of Table 2. Even in these simpler conditions, agents fail in all but one activity (`bringingInWood`, $Q = 0.13$). In a second oracle-driven experiment, we take a successful human demonstration for each activity from the BEHAVIOR Dataset and set a state a few seconds before its successful execution at $T$ as the activity initial state. We train agents with SAC: rows 3 to 6 of Table 2 show the mean success rate ($SR$, full accomplishment of the activity) in 100 evaluation episodes for the final policy resulting from training with three different random seeds ($Q$ starts here close to 1 and is less informative). Even when starting 1 s away from a goal state, most

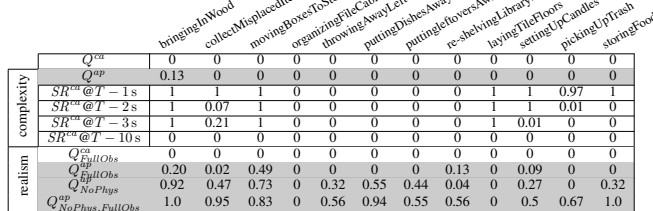

| | | bringingInWood | collectMisplacedItems | movingBoxesToStorage | organizingFileCabinet | throwingAwayLeftovers | puttingDishesAway | puttingLeftoversAway | re-shelvingLibraryBooks | layingTileFloors | settingUpCandles | pickingUpTrash | storingFood |
|---|---|---|---|---|---|---|---|---|---|---|---|---|---|
| **complexity** | $Q^{ca}$ | 0 | 0 | 0 | 0 | 0 | 0 | 0 | 0 | 0 | 0 | 0 | 0 |
| | $Q^{ap}$ | 0.13 | 0 | 0 | 0 | 0 | 0 | 0 | 0 | 0 | 0 | 0 | 0 |
| | $SR^{ca}@T - 1\,s$ | 1 | 1 | 1 | 0 | 0 | 0 | 0 | 0 | 1 | 1 | 0.97 | 1 |
| | $SR^{ca}@T - 2\,s$ | 1 | 0.07 | 1 | 0 | 0 | 0 | 0 | 0 | 1 | 1 | 0.01 | 0 |
| | $SR^{ca}@T - 3\,s$ | 1 | 0.21 | 1 | 0 | 0 | 0 | 0 | 0 | 1 | 0.01 | 0 | 0 |
| | $SR^{ca}@T - 10\,s$ | 0 | 0 | 0 | 0 | 0 | 0 | 0 | 0 | 0 | 0 | 0 | 0 |
| **realism** | $Q^{ca}_{FullObs}$ | 0 | 0 | 0 | 0 | 0 | 0 | 0 | 0 | 0 | 0 | 0 | 0 |
| | $Q^{ap}_{FullObs}$ | 0.20 | 0.02 | 0.49 | 0 | 0 | 0 | 0 | 0.13 | 0 | 0.09 | 0 | 0 |
| | $Q^{ca}_{NoPhys}$ | 0.92 | 0.47 | 0.73 | 0 | 0.32 | 0.55 | 0.44 | 0.04 | 0 | 0.27 | 0 | 0.32 |
| | $Q^{ap}_{NoPhys,FullObs}$ | 1.0 | 0.95 | 0.83 | 0 | 0.56 | 0.94 | 0.55 | 0.56 | 0 | 0.5 | 0.67 | 1.0 |

Table 2: **Evaluation of state-of-the-art RL algorithms on BE-HAVIOR** *Fully realistic, diverse and complex (row 1):* SAC [16] for visuomotor continuous actions (superindex $ca$) performs poorly in all activities; *Complexity analysis (rows 2-6):* reducing complexity (horizon) with temporally extended action primitives (superindex $ap$ and gray cells, trained with PPO [17]) or by starting few seconds away from a goal state, lead to some non-zero success rate ($SR$). *Realism analysis (rows 7-10):* Only by reducing realism in observations and physics, and complexity through action primitives, RL achieves significant success in a handful of the activities.

agents fail. A few do better, but their performance decreases quickly as we start further away from the successful execution, being zero for all activities at $10\,\mathrm{s}$. This indicates that the long horizon of BEHAVIOR activities is in fact a paramount challenge for RL. We hypothesize that Embodied AI solutions with a hierarchical structure such as hierarchical-RL or task-and-motion-planning (TAMP) may help to overcome the challenges of high complexity (length) of the BEHAVIOR activities [69–72].

**Effect of realism (in sensing and actuation):** In a third experiment, we evaluate how much the realism in actuation and sensing affects the performance of embodied AI solutions. We train agents with continuous motion control (SAC), and motion primitives (PPO) assuming full-observability of the state, with results in Tables 2 (rows 7-8, subindex *FullObs*). Even with full observability, the complexity dominates policies in the original action space and they fail entirely. For policies selecting among action primitives, there is partial success in only five activities, indicating that perception is part of the difficulty in BEHAVIOR. To evaluate the effect of realistic actuation, we train an agent using action primitives that execute without physics simulation, achieving their expected outcome (e.g. grasp an object, or place it somewhere). Tables 2 (row 9-10, subindex *noPhys*) shows the results, also in combination with unrealistic full-observability. We observe that without the difficulties of realistic physics and actuation, the learning agents achieve an important part of most activities, accomplishing consistently two of them ($Q = 1$) when full-observability of the state is also granted. This indicates that the generation of the correct actuation is a critical challenge for embodied AI solutions, even when they infer the right next step at the task-planning level, supporting the importance of benchmarks with realistically action execution over predefined action outcomes.

**Effect of diversity (in activity instance and objects):** Another cause of the poor performance of robot learning solutions in the 12 BEHAVIOR activities may be the high diversity in multiple dimensions, such as scenes, objects, and initial states. This diversity forces embodied AI solutions to generalize to all possible conditions. In a second experiment,

| Diversity in… | | | | | | |
|---|---|---|---|---|---|---|
| object poses | object instances | ontop | sliced | soaked | stained | cooked |
| ✖ | ✖ | 1 | 0.15 | 1 | 1 | 1 |
| ✔ | ✖ | 0.825 | 0 | 0.935 | 0.28 | 0.66 |
| ✔ | ✔ | 0.46 | 0 | 0.925 | 0.11 | 0.265 |

Table 3: **Evaluation of the effect of BEHAVIOR's diversity:** Results of training agents with SAC [16] in single-predicate activities of increasing diversity; Even in these simple activities, performance degrades quickly indicating that current state-of-the-art cannot cope with the dimensions of diversity spanned in BEHAVIOR

we evaluate the effect of BEHAVIOR's diversity on performance. To present diversity across activities while alleviating their complexity, we train RL agents to complete five single-literal activities involving only one or two objects. Note that these activities are not part of BEHAVIOR. We evaluate training with RL (SAC) for each activity under diverse instantiations: initialization of the activity (object poses) and object instances. The results are shown in Table 3, where we report $Q$. First, we train without any diversity as baseline to understand the ground complexity of the single-literal activities. All agents achieve success. Then, we evaluate how well the RL policies train for a diverse set of instances of the activities, first changing objects' initial pose, then changing the object. Performance in all activities decreases rapidly, especially in sliced and stained. These experiments indicate that the diversity in BEHAVIOR goes beyond what current RL algorithms can handle even in simplified activities, and poses a challenge for generalization in embodied AI.

# 8 Conclusion and Future Work

We presented BEHAVIOR, a novel benchmark for embodied AI solutions of household activities. BEHAVIOR presents 100 realistic, diverse and complex activities with a new logic-symbolic representation, a fully functional simulation-based implementation, and a set of human-centric metrics based on the performance of humans on the same activities in VR. The activities push the state-of-the-art in benchmarking adding new types of state changes that the agent needs to be able to cause, such as cleaning surfaces or changing object temperatures. Our experiments with two state-of-the-art RL baselines shed light on the challenges presented by BEHAVIOR's level of realism, diversity and complexity. BEHAVIOR will be open-source and free to use; we hope it facilitates participation and fair access to research tools, and paves the way towards a new generation of embodied AI.

**Acknowledgments**

We would like to thank Bokui Shen, Xi Jia Zhou, Jim Fan, Manasi Sharma, and Sidhart Krishnan for comments, ideas, and support in data collection. This work is in part supported by Toyota Research Institute (TRI), ARMY MURI grant W911NF-15-1-0479, Samsung, Amazon, and Stanford Institute for Human-Centered AI (SUHAI). S. S. and C. L. are supported by SUHAI Award #202521. S. S. is also supported by the National Science Foundation Graduate Research Fellowship Program (NSF GRFP). R. M-M. and S. B. are supported by SAIL TRI Center – Award # S-2018-28-Savarese-Robot-Learn. S. B. is also supported by a National Defense Science and Engineering Graduate (NDSEG) fellowship, SAIL TRI Center – Award # S-2018-27-Niebles, and SAIL TRI Center – Award # TRI Code 44. S. S. and S. B. are supported by a Department of Navy award (N00014-16-1-2127) issued by the Office of Naval Research (ONR). F. X. is supported by the Qualcomm Innovation Fellowship and Stanford Graduate Fellowship. This article solely reflects the opinions and conclusions of its authors and not any other entity.

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
