# OpenReview forum: "BEHAVIOR: Benchmark for Everyday Household Activities in Virtual, Interactive, and Ecological Environments"
_robot-learning.org/CoRL/2021/Conference — CoRL2021 Poster_

### Official Review · Reviewer_BvZu · 2021-07-06

**Originality:** Very Good
**Technical Quality:** Good
**Clarity Of Presentation:** Good
**Impact:** 4

**Recommendation:**

Strong Accept: I recommend accepting the paper and will argue for my recommendation even if other reviewers hold a different opinion.

**Summary:**

The paper introduces a simulator-agnostic predicate logic framework for defining task specifications and goal completion, as well as associated metrics. The paper demonstrates an implementation in iGibson for 100 human activities, and presents a dataset of 500 human demonstrations of such activities gathered via virtual reality with a direct mapping to simulated agent control.
The collection of human demonstrations that can be replayed, understood, and evaluated in simulation for training agent models is the primary contribution of this paper.

*Post-response*
The authors did a nice job clarifying some questions I and other reviewers raised. The paper contributions are still "too big" to be squeezed into 8 pages, but with the supplementary material available I think that's fine. The audience at CoRL will find the dataset useful, and if the tooling is easy to use off the shelf, the data may spawn a lot of follow up methods and approaches that will be fun to see.

**Issues:**

Questions in order of importance

Q1: What is the motivation for using sample-inefficient RL techniques only in section 7 for evaluation? With human-level demonstrations via VR, doesn't teacher forcing stand out as a cleaner and more straightforward starting point? It's somewhat unsurprising that performance is zero (Table 2 row 1, most of row 2) for RL.

Q2: While full physics simulation is touted as an important part of the instantiation for the dataset, section 5 indicates the action space is discretized into 6 actions which take object targets directly [e.g., 'open fridge']. Why use a full physics simulation if inference will be reduced to a semantics-level action space?

Q3: For Efficiency metrics, why is Displacement of Hands calculated only while in contact with an object? In particular, that metric will miss the planning time and efficiency needed for grasping the object / planning to make initial contact.

Nits (not effecting evaluation):

- The backronym title is too painfully forced; I feel like you'll mostly get backlash in reception for the awkward "O" and "R".

- Line 28 stray right paren

- Line 33 "to accommodate" -> "from accommodating"

- Line 52 "comprise of" -> "comprise" or "are composed of"

- Line 89 no closing parent for open left paren

- The close similarity of using BDDL to instantiate possible environments with ALFRED's use of PDDL to instantiate environments and create automated plans towards goal conditions could be mentioned.

**Reviewer Expertise:**

Very good: Comprehensive knowledge of the area

**Strengths And Weaknesses:**

The main strength of the paper is the contribution of human-level, VR-based task completion demonstrations.
That real-world activities curated by a third party are used is a strength that could be further emphasized in the writing. For example, the decision to use real-world activities brings this paper's data demonstrations closer in line to "Ecologically Valid" AI.

@inproceedings{devries:arxiv20,
  author = {Harm de Vries and Dzmitry Bahdanau and Christopher Manning},
  booktitle = {arXiv},
  title = {Towards Ecologically Valid Research on Language User Interfaces},
  year = {2020},
}

The main weakness of the paper is that CoRL isn't the "perfect" fit venue for it. The contributions are substantial in both data and implementation, but are limited to simulation, while CoRL as a space prefers actual robot learning.

**Summary Of Recommendation:**

The VR dataset and benchmark implementation in iGibson is substantial. While CoRL may not be the perfect venue for such a benchmark since it's currently fully in simulation, I feel the benchmark takes meaningful steps towards "Embodied AI" and eventual physical agent transfer.

---

> ### Author Response · Authors · 2021-08-31
> **Response to reviewer BvZu (3/3)**
>
> > **Q6**: "Comparison BDDL with PDDL in ALFRED"
>
> **A6**: Regarding the use of PDDL in ALFRED: Thank you for noting this - we agree that it’s important to explain the uniqueness of BDDL. The main common ground is that the languages have very similar structure of initial and goal conditions. There are also some key distinctions: like other uses of PDDL, ALFRED’s logical operators are a subset of BDDL’s, with BDDL’s additional operators such as ForPairs, ForN, and ForNPairs (formalized in sec. A.3.2) being designed for ease of use. These only obey first-order logic axioms for certain object spaces, but help annotators unfamiliar with formal logic write flexible definitions. For example, *ForPairs: apple, bag. inside(apple, bag)* says that some one-to-one pairing of apples and bags must be made where each apple is in its corresponding bag. This is more general than saying e.g. “inside(apple1, bag1) AND inside(apple2, bag2)....” and helps annotators avoid such a brittle scenario. Another difference is that ALFRED’s use for PDDL is to annotate action plans from demonstrations and use planners much like in other applications of PDDL, whereas BDDL does not specify plans and describes only world states. This speaks to a fundamental difference in the purpose of having a domain-specific language, as we see BDDL as a usable, flexible, and extensible way to define naturalistic activities beyond its uses in planning. Please see the additions in sec. A.3.2 for details explaining this and a more general comparison of BDDL and various PDDLs.
>
> ### References
> - [1] Xia, F., Li, C., Martín-Martín, R., Litany, O., Toshev, A., & Savarese, S. (2021). ReLMoGen: Leveraging motion generation in reinforcement learning for mobile manipulation. In Proceedings of the International Conference on Robotics and Automation (ICRA)
> - [2] Zhou, M., Luo, J., Villella, J., Yang, Y., Rusu, D., Miao, J., ... & Wang, J. (2020). Smarts: Scalable multi-agent reinforcement learning training school for autonomous driving. Proceedings of the 4th Conference on Robot Learning (CoRL).
> - [3] Kolve, E., Mottaghi, R., Han, W., VanderBilt, E., Weihs, L., Herrasti, A., ... & Farhadi, A. (2017). Ai2-thor: An interactive 3d environment for visual ai. arXiv preprint arXiv:1712.05474.

---

> ### Author Response · Authors · 2021-08-31
> **Response to reviewer BvZu (2/3)**
>
> >**Q3**: “For Efficiency metrics, why is Displacement of Hands calculated only while in contact with an object? In particular, that metric will miss the planning time and efficiency needed for grasping the object / planning to make initial contact.”
>
> **A3**: That is right, the pregrasp (and other hand motion) is not considered in our metric.  Our metric tries to approximate the effort of the agent when and for interacting with objects. We are interested in knowing how much end-effector motion is used by humans to manipulate.
> We observed that humans move their hands considerably even when not contacting objects (not for manipulation). For example, when walking or moving around. Initially, we considered a metric that aggregated ALL hand motion but in this metric the actual motion of the hands for manipulation is buried in the large amount of random motion of the hands while moving around.
>
> >**C4**: “The main weakness of the paper is that CoRL isn't the "perfect" fit venue for it. The contributions are substantial in both data and implementation, but are limited to simulation, while CoRL as a space prefers actual robot learning.”
>
> **A4**: We completely agree with the reviewer on the importance of real robot experiments for robotics research. We also believe that our submission on BEHAVIOR is a good fit for the conference. This is because our goal with BEHAVIOR is to provide a highly realistic challenge in simulation to foster research and solutions that can ultimately be leveraged to real robots in the real world, possibly with some domain adaptation. That’s why we use a realistic physics engine, realistic object models, and implement the model of a realistic mobile manipulator (Fetch). The underlying simulator, iGibson, and the physics engine, Bullet, have been used in prior work to develop solutions for real world problems. The simulated virtual sensor signals are realistic, as the comparison in the appendix (page 16) of this paper using iGibson shows [1].
>
> Our immediate next steps involve evaluating some of our findings in BEHAVIOR in the real world, now that we have regained access to the lab and the real robot. We hope to compare both human and robot performance in real world versions of our activities.
> The challenge in BEHAVIOR is to develop an AI solution that learns to control physical interactions based on onboard sensor signals in a realistically simulated robot. While simulation is not the real world, it is a great tool to develop solutions for the real world, as it was demonstrated by last year’s CoRL Best System Paper award “SMARTS: Scalable Multi-Agent Reinforcement Learning Training School for Autonomous Driving” by Zhou et al., a simulation environment to study and develop solutions for autonomous cars [2].
>
> >**C5**: Response to “nits”, including grammatical errors
>
> **A5**: Thank you for the excellent proofreading - we have addressed all your syntax notes!

---

> ### Author Response · Authors · 2021-08-31
> **Response to reviewer BvZu (1/3)**
>
> Thanks for the constructive feedback and we provide responses as below:
>
> >**Q1**: “What is the motivation for using sample-inefficient RL techniques only in section 7 for evaluation? With human-level demonstrations via VR, doesn't teacher forcing stand out as a cleaner and more straightforward starting point? It's somewhat unsurprising that performance is zero (Table 2 row 1, most of row 2) for RL.”
>
> **A1**: We agree that it is unsurprising that standard model-free reinforcement learning methods fail to complete an entire BEHAVIOR task. Our goal with this benchmark is to establish a new level of task complexity, on the axes of realism, diversity, and complexity. We believe that this benchmark necessitates the development of novel methods to show meaningful success, however, it is important to show that prior state-of-the-art methods are insufficient to solve the challenge. We believe a promising future direction is leveraging the task-plan, extracted from human demonstrations, to use in combination with our motion primitives to solve the activities. We hope to post results showing the viability of this method before the deadline.
>
> >**Q2**: While full physics simulation is touted as an important part of the instantiation for the dataset, section 5 indicates the action space is discretized into 6 actions which take object targets directly [e.g., 'open fridge']. Why use a full physics simulation if inference will be reduced to a semantics-level action space?
>
> **A2**: This is an important question that we may need to make more clear in the text. In the text, we indicate that we “additionally” provide action primitives implemented with motion planning techniques. First, the execution of these primitives includes a full physical simulation (except in the ablation study where we “deactivate” physics to evaluate its effect). That means that the execution of the primitives can still fail due to realistic physical interactions. This is very different from a pure semantics-level action space with pre and post-conditions like in AI2-THOR[3], where “if fridge is close AND fridge is atReach'' the outcome of the semantic-level action “open fridge” is directly the fridge in an open state. Only in one ablation study do we “deactivate physics”, and only to evaluate how much the performance improves, to gain information about how hard it is to control the realistically simulated physics. Second, these primitives are only to facilitate the development of solutions for researchers interested mostly in the task-level planning, not on the control of the interaction, and that can live with noisy and failure prone low level primitives. But, as mentioned, the primitives can fail and they are also not complete: e.g, there is no primitive to call to clean the entire floor, or to slice fruit. The primitives are thus an additional tool we provide, but not an alternative action space.
>
> In general, we see the primitives as (possibly) part of the solution to the tasks. In BEHAVIOR, the problem we define is, as it is the case in the real world, to control the low-level actions (joint motion, end-effector motion, base motion, head motion) of an embodied agent/robot based on continuously arriving signals from the onboard sensors (cameras, encoders, …). Some researchers believe that the best solution is to create primitives as a middle layer between planning and control of the agent. To those researchers, we provide some initial set of primitives that they can use, modify or extend. Some others do not believe that primitives are necessary, or they may want to create their own primitives (e.g., learned policies). All paths are valid, as far as there are control signals to the agent at 30 fps, with our provided primitives, other primitives (e.g. learned policies) or no primitives at all.

---

> ### Author Response · Authors · 2021-08-31
> **Follow-up response**
>
> Dear Reviewer:
>
> We apologize for the late post, we have been working hard to obtain empirical results that will clarify some of the reviewers' comments. These two additional replies relate to general questions about our submission:
>
> **On the use of VR data as imitation data to develop AI solutions:** As referenced in our earlier review, we have conducted an experiment to evaluate the utility of VR demonstrations to develop AI solutions. Using the logical predicate checking functions, we were able to extract kinematic and non-kinematic state changes for each human demonstration in order to segment the demonstrations into task plans. Segments correspond to some of the action primitives we have developed and provided as support for other researchers. For example, the “re-shelving library book” activity will have a task plan that looks like this: NavigateTo(book), Grasp(book), PlaceInside(book, shelf), etc. We then implemented partially simulated action primitives (see Appendix A.4) that allowed the agent to evaluate this task plan in the BEHAVIOR benchmark, and simulate the effect of the actions. Our agent successfully achieved 27 of the 238 task demos replayed, showing partial success in 85 of them. A partial success indicates that the agent successfully changed to True at least one of the predicates of the goal condition. Over all 238 demos, the agent switches to True 20% of the False predicates in the goal conditions.
>
> This is a promising first step to use human demonstrations as a source to learn high level plans. If the agent equipped with basic primitives is able to reproduce some of the success of the human agent, the VR demonstrations can be processed in this manner to learn to produce high-level plans. Future development in this line of research will need to improve the generation of task-plans from direct cloning them to learning to generate plans based on task conditions, and improve (possibly with data driven methods) the set of action primitives. Alternatively, these preliminary results should not discourage exploring other IL approaches that use the raw motion and interaction data in the VR demonstrations, without any intermediate action primitives; however, we believe this will be harder as the number of demonstrations per activity may not be not large enough to learn complex motion primitives.
>
> **On realistic interactions (using a realistic robot model)**: In addition, we would like to further illustrate the use of the simulated Fetch robot in BEHAVIOR. We have recorded a video of a teleoperator controlling the Fetch model. The user performs the "cleaning_table_after_clearing" BEHAVIOR activity to illustrate the use of a realistic robot model in BEHAVIOR tasks:
>
> https://streamable.com/pj7nu0
>
> While it is more difficult to grasp with a single fetch gripper, as can be seen in the video, the user is able to navigate through doors, pick up objects, trigger multiple non-kinematic state changes (toggling on the faucet and soaking the cloth). In this video, grasping is fully simulated with rigid body contacts between the two fingers and the objects. This embodiment is fully functional and available for the benchmark, modeled as close as possible to the real robot to provide a more realistic interaction and strategies that are close to the ones required by real robots.
>
> We hope this further clarifies some of the questions and comments by the reviewers.
>
> Best regards,
>
> Authors of CoRL 2021 Conference Paper87

---

### Official Review · Reviewer_GSeR · 2021-07-23

**Originality:** Very Good
**Technical Quality:** Very Good
**Clarity Of Presentation:** Very Good
**Impact:** 4

**Recommendation:**

Weak Accept: I recommend accepting the paper, but will not argue for my recommendation if the majority of other reviewers have a different opinion.

**Summary:**

This paper develops a realistic, diverse and complex benchmark with 100 household activities in simulation for embodied AI, aiming to reproduce the challenges that embodied agents must face in real world. To this end, the authors first propose a predicate logic-based language (BDDL) to enabling general definition of activities and unlimited generation of diverse instances of each activity. Besides, the authors claimed that BEHAVIOR is designed to be not bounded to any specific simulator and can be deployed in simulators that satisfy some functional requirements. Further, the authors propose a primary metric based on success and several secondary metrics for characterizing efficiency. The BEHAVIOR includes VR interfaces to collect human demonstration.
In the experiments, the authors show that: 1. the full BEHAVIOR benchmark is challenging 2. Realism, diversity and complexity indeed contributes to task-difficulty respectively.


**Issues:**

1.	I hope the authors can address my concerns in ‘Weaknesses’ part.
2.	The proposed benchmark seems to be the most challenging tasks I have ever seen, what do you think is the promising-direction/key-aspect to solve this problem in the future?
3.	There are built-in agents in recent embodied AI benchmarks [1] [2]. Do you consider develop built-in agent that can solve these tasks with high success rate?

[1] L. Weihs, M. Deitke, A. Kembhavi, and R. Mottaghi. Visual room rearrangement. arXiv preprint arXiv:2103.16544, 2021.
[2] Puig, X., Shu, T., Li, S., Wang, Z., Tenenbaum, J. B., Fidler, S., and Torralba, A. Watch-And-Help: A Challenge for Social Perception and Human-AI Collaboration. arXiv preprint arXiv:2010.09890, 2020.


**Reviewer Expertise:**

Good: General knowledge of the area

**Strengths And Weaknesses:**

Strengths:
1.	The paper is clearly-written and well-organized.
2.	The benchmark proposed in this paper requires heavy engineering to implement.
3.	The authors put forward a very challenging and meaningful benchmark, which I believe will be very helpful to the community in the future.
4.	The benchmark takes the advantage of human demonstration(VR interfaces), which is novel and helpful.
5.	The benchmark is not bounded to a specific simulator, and the authors identify the simulator-agnostic features required for deploying the benchmark, which I think this will be the trend in the future.

Weaknesses:
1.	The method evaluated in “Results in the original activities” experiment is too simple. I think authors should try to implement a more advanced end2end pipeline in embodied AI literature just as [1] does.

2.	The human demonstrations collected by VR is not evaluated in experiments. I really appreciate the idea that the platform can take advantage of human display, but I also want to see experiments prove that this idea is very helpful for challenging tasks

[1] L. Weihs, M. Deitke, A. Kembhavi, and R. Mottaghi. Visual room rearrangement. arXiv preprint arXiv:2103.16544, 2021.


**Summary Of Recommendation:**

The proposed benchmark is meaningful and challenging. Some key ideas involved in this benchmark is thoughtful: the task definition should be comprehensive and enable unlimited instances generation, the benchmark should be simulator-agnostic and we should take advantage of human demonstrations. It will be better if authors evaluate more advanced method on this benchmark and evaluate the human demonstrations, to address the concerns I mentioned in the ‘Weakness’ part.

---

> ### Author Response · Authors · 2021-08-31
> **Response to Reviewer GSeR (2/2)**
>
> ### References
> - [1] Haarnoja, T., Zhou, A., Abbeel, P., & Levine, S. (2018, July). Soft actor-critic: Off-policy maximum entropy deep reinforcement learning with a stochastic actor. In International conference on machine learning (pp. 1861-1870). PMLR.
> - [2] Haarnoja, T., Zhou, A., Hartikainen, K., Tucker, G., Ha, S., Tan, J., ... & Levine, S. (2018). Soft actor-critic algorithms and applications. arXiv preprint arXiv:1812.05905.
> - [3] Wahid, A., Stone, A., Chen, K., Ichter, B., & Toshev, A. (2020). Learning object-conditioned exploration using distributed soft actor critic. Conference on Robot Learning.
> - [4] Weihs, L., Deitke, M., Kembhavi, A., & Mottaghi, R. (2021). Visual Room Rearrangement. In Proceedings of the IEEE/CVF Conference on Computer Vision and Pattern Recognition (pp. 5922-5931).

---

> ### Author Response · Authors · 2021-08-31
> **Response to Reviewer GSeR (1/2)**
>
>
> Thanks for the constructive feedback and we provide responses as below:
>
> > **C1**: “The method evaluated in “Results in the original activities” experiment is too simple”
>
> **A1**: The method in our experimental setup is a standard SAC implementation that has performed well in other domains [1,2,3]. Therefore, we believe it is valuable to measure its performance and perform ablations to understand why a state-of-the-art model-free RL algorithm suffers in the BEHAVIOR tasks.
> The cited Visual Room Rearrangement paper by Weihs et al. [4] presents an excellent approach to their task and simulator conditions. We tried to find the implementation of their solution but we couldn’t find it online. In any case, their method assumes a different action space with discrete actions. It would be necessary to make a significant effort to modify their approach for the BEHAVIOR tasks.
> In general, Visual Room Rearrangement [4] has a heavier load on the solution: they present only one task (taking one object and bringing it to a desired location) and an advanced solution to it. In our paper, the focus is different: we present an exhaustive set of 100 activities, fully instantiated in a simulator, defined with a extensible logic-semantic language, with multiple VR demonstrations for all of them, and the evaluation of a vanilla state-of-the-art algorithm to understand the complexity of the activities. We are actively working on multiple solutions to the very complex activities we present in BEHAVIOR. As reviewer NSMD mentioned, our paper currently contains a significant amount of information with a simple RL algorithm in the evaluation; including a novel and complex algorithm together with the explanation and analysis of the benchmark would be hard in a single paper.
>
> > **C2**: “The human demonstrations collected by VR is not evaluated in experiments”
>
> **A2**: Our main goal with the VR demonstrations is to use them as relative measures to understand the performance of AI agents compared to humans. We have used the VR demonstrations to establish baselines for human level performance across the various metrics proposed and implemented for the BEHAVIOR benchmark. However, we assume that “evaluate VR in experiments” means demonstrate that the VR demonstrations can be used to train policies. Please, if this is not the intended meaning, let us know. We truly believe the demonstrations contain useful information, not only to be used in the metrics, but also to train agents. As mentioned to reviewers NSMD and Ljtb, we are working on an imitation learning solution that makes use of the demonstrations to provide high level task plans that the agent executes with its own implementation of the provided set of low level primitives.
>
> > **Q3**: “The proposed benchmark seems to be the most challenging tasks I have ever seen, what do you think is the promising-direction/key-aspect to solve this problem in the future?”
>
> **A3**: We also believe that these tasks are extremely challenging and they are going to require a significant effort from the community to be solved. We think that the increased difficulty in the three axes we identified (realism, diversity and complexity) will require new types of solutions. Concretely, it will be hard to train adhoc solutions since the scenes, objects and initial states change between instances of the same activity. Generalizing through object semantics is a way to overcome these challenges of diversity. We also think that some of the difficulties in the BEHAVIOR tasks come from the very long horizon of the activities, as shown in our experiments. For these very long tasks, some form of hierarchical planning and abstraction will be necessary, e.g., task and motion planning or some form of hierarchical imitation/reinforcement learning. Finally, we believe that another difficulty in BEHAVIOR is caused by the mobile manipulation setup: robots that can navigate and interact in large spaces introduce additional challenges like exploration of the environment and partial observability. Some form of memory mechanism may be required to cope with these problems.
>
> > **Q4**: “There are built-in agents in recent embodied AI benchmarks. Do you consider develop built-in agent that can solve these tasks with high success rate?”
>
> **A4**: We are actively developing different approaches to improve the (poor) performance of model-free RL in BEHAVIOR. All our solutions will be part of a set of built-in baseline agents for others to use as templates and comparisons. Some of the solutions we are developing include imitation learning, task planning and skill learning, and motion planning. We hope this will help to build increasingly better solutions and compare to previous ones.

---

> ### Author Response · Authors · 2021-08-31
> **Follow-up response**
>
> Dear Reviewer:
>
> We apologize for the late post, we have been working hard to obtain empirical results that will clarify some of the reviewers' comments. These two additional replies relate to general questions about our submission:
>
> **On the use of VR data as imitation data to develop AI solutions:** As referenced in our earlier review, we have conducted an experiment to evaluate the utility of VR demonstrations to develop AI solutions. Using the logical predicate checking functions, we were able to extract kinematic and non-kinematic state changes for each human demonstration in order to segment the demonstrations into task plans. Segments correspond to some of the action primitives we have developed and provided as support for other researchers. For example, the “re-shelving library book” activity will have a task plan that looks like this: NavigateTo(book), Grasp(book), PlaceInside(book, shelf), etc. We then implemented partially simulated action primitives (see Appendix A.4) that allowed the agent to evaluate this task plan in the BEHAVIOR benchmark, and simulate the effect of the actions. Our agent successfully achieved 27 of the 238 task demos replayed, showing partial success in 85 of them. A partial success indicates that the agent successfully changed to True at least one of the predicates of the goal condition. Over all 238 demos, the agent switches to True 20% of the False predicates in the goal conditions.
>
> This is a promising first step to use human demonstrations as a source to learn high level plans. If the agent equipped with basic primitives is able to reproduce some of the success of the human agent, the VR demonstrations can be processed in this manner to learn to produce high-level plans. Future development in this line of research will need to improve the generation of task-plans from direct cloning them to learning to generate plans based on task conditions, and improve (possibly with data driven methods) the set of action primitives. Alternatively, these preliminary results should not discourage exploring other IL approaches that use the raw motion and interaction data in the VR demonstrations, without any intermediate action primitives; however, we believe this will be harder as the number of demonstrations per activity may not be not large enough to learn complex motion primitives.
>
> **On realistic interactions (using a realistic robot model)**: In addition, we would like to further illustrate the use of the simulated Fetch robot in BEHAVIOR. We have recorded a video of a teleoperator controlling the Fetch model. The user performs the "cleaning_table_after_clearing" BEHAVIOR activity to illustrate the use of a realistic robot model in BEHAVIOR tasks:
>
> https://streamable.com/pj7nu0
>
> While it is more difficult to grasp with a single fetch gripper, as can be seen in the video, the user is able to navigate through doors, pick up objects, trigger multiple non-kinematic state changes (toggling on the faucet and soaking the cloth). In this video, grasping is fully simulated with rigid body contacts between the two fingers and the objects. This embodiment is fully functional and available for the benchmark, modeled as close as possible to the real robot to provide a more realistic interaction and strategies that are close to the ones required by real robots.
>
> We hope this further clarifies some of the questions and comments by the reviewers.
>
> Best regards,
>
> Authors of CoRL 2021 Conference Paper87

---

### Official Review · Reviewer_Ljtb · 2021-07-23

**Originality:** Very Good
**Technical Quality:** Very Good
**Clarity Of Presentation:** Excellent
**Impact:** 4

**Recommendation:**

Weak Accept: I recommend accepting the paper, but will not argue for my recommendation if the majority of other reviewers have a different opinion.

**Summary:**

The paper proposes the BEHAVIOR benchmark for simulated household tasks for embodied AI agents. It builds on the iGibson2.0 simulator and includes 100 long-horizon indoor manipulation tasks. The paper defines a high-level PDDL-style language for defining tasks which enables the generation of arbitrary many instances of a task, and further introduces new datasets of teleoperated human demonstrations for the 100 tasks.


**Issues:**

I hope the authors can clarify some of my questions about the realism of the simulated tasks on the lower level and also clarify what kind of manipulations are required to solve the tasks. I strongly believe that the benchmark could benefit from additional robot demonstrations of some form to increase its usability and give hope that progress is possible, but I don't think it's strictly necessary for it to be a meaningful contribution.


**Reviewer Expertise:**

Very good: Comprehensive knowledge of the area

**Strengths And Weaknesses:**

## Strengths
- the benchmark features a substantially more diverse set of long-horizon tasks than any existing embodied AI benchmark --> this enables research on more challenging problems

- the PDDL-style task definition language allows to sample arbitrary many instances of a task that differ in object arrangement / scene layout etc --> this enables research on more robust / generalizable problem solving systems

- the paper provides a comprehensive overview of prior related benchmarks and clearly delineates the differences

- the paper provides human demonstrations for all tasks and defines a way to use them for evaluation of learning system that goes beyond raw goal-state reaching towards measuring whether a task was solved "in a reasonable way". I am not aware of prior benchmarks that evaluated like this (but I might be missing some)

- the paper is well written and easy to follow

- the paper provides a clear separation between the task-side of the benchmark and the simulator used for implementing it


## Weaknesses
My main concern is that the paper does not clearly explain the realism of the required interactions for solving tasks. There are many places in the paper that mention the realism of the benchmark, but they refer more to the high-level diversity of tasks, which I agree is more realistic than in existing benchmarks. But it lacks a clear description of what level of realism is present in the execution of each of the subtasks. Yet, this is crucial to understand the challenges posed to learning systems and how this benchmark differs from others.

In particular the paper leaves questions open about the realism on two levels:
(1) on the very low-level: how realistically are interactions eg between the agent and objects simulated. Reading through the attached iGibson2.0 paper I saw that it uses an "assisted grasping" mechanism that can remove a lot of the subtle complexity of realistic object interactions (contact forces, slipping etc). While I understand why such compromises need to be made I would like to (a) see them mentioned in the benchmark paper clearly to show what challenges the benchmark poses, including in Tab.1 where other benchmarks potentially feature more realistic object interaction physics etc and (b) understand whether such abstractions are also used for the robot experiments, eg does the robot also use assisted grasping for object interactions?

(2) on a higher level the paper lacks details what it means to "achieve" certain tasks. Eg what is required to "load the dishwasher" or to "fill the stockings"? How precise do I need to control the robot? Adding some examples of this in the main text and a more detailed account in the appendix could help understand the difficulty of the lower-level manipulations required for the tasks.

Similarly, it is unclear to me from reading the paper how much different the low-level manipulations required are from eg object rearrangement. I agree that the diversity of high-level tasks is much higher, but if all of these tasks just require moving the EEF close to object 1, then calling discrete grasp action A, then moving close to location 2 and calling discrete release action B, they are not a lot more diverse than rearrangement in terms of the low-level manipulations. More clarity on point (2) above could help to clarify this. (A discussion on such abstractions eg in grasping should probably also be added to the sim2real discussion in the appendix.)

Another limitation is the infeasibility of the tasks for current RL methods. Given the limited success of methods in all but the simplest settings, it is likely that some form of demonstration data is needed to make meaningful progress on these tasks. The authors mention that the human data could be used to help learning, but I am skeptical of the practicality of this due to the large disparity in embodiment. Instead the usability of the benchmark could increase a lot if robot demonstrations were provided, either by teleoperation or by providing scripted demonstrations. The latter might not be too hard to achieve given that the benchmark already comes with implemented action primitives.


## Questions
(A) Can users define new tasks in the framework? If yes, what does this take?

(B) What does it mean to provide two example definitions per task? (L126)

(C) How can we compare different solutions in the efficiency metric? We can probably trade one for the other, eg disarrange the environment more but minimize time / path length by knocking over a vase etc. Disarranging a chair is also a lot less severe than disarranging a vase by braking it etc. I am wondering whether the authors have thoughts on how to actually use these qualitative metrics for comparison / evaluation.

**Summary Of Recommendation:**

I am excited about this benchmark. If the authors can address some of the remaining unclear points I raised above and include their discussion in the paper, I am happy to recommend acceptance.

---

> ### Author Response · Authors · 2021-08-31
> **Response to Reviewer Ljtb (5/5)**
>
> ### References
> - [1] Iscen, A., Caluwaerts, K., Tan, J., Zhang, T., Coumans, E., Sindhwani, V., & Vanhoucke, V. (2018, October). Policies modulating trajectory generators. In Conference on Robot Learning (pp. 916-926). PMLR.
> - [2] Peng, X. B., Coumans, E., Zhang, T., Lee, T. W., Tan, J., & Levine, S. (2020). Learning agile robotic locomotion skills by imitating animals. arXiv preprint arXiv:2004.00784.
> - [3] Yu, W., Tan, J., Bai, Y., Coumans, E., & Ha, S. (2020). Learning fast adaptation with meta strategy optimization. IEEE Robotics and Automation Letters, 5(2), 2950-2957.
> - [4] Seita, D., Florence, P., Tompson, J., Coumans, E., Sindhwani, V., Goldberg, K., & Zeng, A. (2021). Learning to Rearrange Deformable Cables, Fabrics, and Bags with Goal-Conditioned Transporter Networks. IEEE International Conference on Robotics and Automation (ICRA).
> - [5] Mason, M. T. (1981). Compliance and force control for computer controlled manipulators. IEEE Transactions on Systems, Man, and Cybernetics, 11(6), 418-432.
> - [6] Khatib, O. (1987). A unified approach for motion and force control of robot manipulators: The operational space formulation. IEEE Journal on Robotics and Automation, 3(1), 43-53.
> - [7] James, S., Ma, Z., Arrojo, D. R., & Davison, A. J. (2020). Rlbench: The robot learning benchmark & learning environment. IEEE Robotics and Automation Letters, 5(2), 3019-3026.
> - [8] Lin, X., Wang, Y., Olkin, J., & Held, D. (2020). Softgym: Benchmarking deep reinforcement learning for deformable object manipulation. In Conference on Robot Learning. PMLR
> - [9] Yu, T., Quillen, D., He, Z., Julian, R., Hausman, K., Finn, C., & Levine, S. (2020, May). Meta-world: A benchmark and evaluation for multi-task and meta reinforcement learning. In Conference on Robot Learning (pp. 1094-1100). PMLR.
> - [10] Hejna, D., Pinto, L., & Abbeel, P. (2020, November). Hierarchically decoupled imitation for morphological transfer. In International Conference on Machine Learning (pp. 4159-4171). PMLR.
> - [11] Zhang, Q., Xiao, T., Efros, A. A., Pinto, L., & Wang, X. (2021). Learning Cross-Domain Correspondence for Control with Dynamics Cycle-Consistency. International Conference on Learning Representations.

---

> > ### Comment · Reviewer_Ljtb · 2021-09-03
> > **Rebuttal Response**
> >
> > Thanks for your thorough response! It confirmed my positive impression in the first review and I support acceptance of the submission.
> >
> > I totally understand why the assistive grasping is used as part of the human VR interface and I hope that the camera ready version of the paper will include this argumentation. I also found the concrete examples of the task complexity and the video of one of the "non-rearrangement-style" tasks very helpful for understanding scope and complexity of tasks. Please do include more such examples in the camera ready!

---

> ### Author Response · Authors · 2021-08-31
> **Response to Reviewer Ljtb (4/5)**
>
> > **Q5**: “Can users define new tasks in the framework? If yes, what does this take?”
>
> **A5**: Yes, BDDL is designed with usability in mind! The user needs to write a syntactically correct BDDL problem file, meaning they define 1) an object space using only supported categories, 2) an initial condition, and 3) a goal condition (sec. A.3.2 lists content constraints and guidelines, e.g. the initial condition can only have constant terms).
>
> Given a syntactically correct BDDL problem and a scene, the BDDL implementation (available after reviewing period) and sampling functionality specified in the paper and implemented in iGibson 2.0 can create infinite satisfactory instances. If the initial or goal condition is physically incompatible with the given scene (e.g. asking for 30 large watermelons in one small cupboard), the sampling functionality gives readable feedback.
> To avoid syntax errors, misattributed properties, etc., users can write definitions with our annotation interface (sec. A.3.3, fig. A.4). This provides a visually intuitive interface that guarantees correct syntax, supported categories, and supported category-property pairing. Both an online server to generate task definitions as well as the code to create and/or modify another server will be available after the reviewing process.
>
> > **Q6**: “What does it mean to provide two example definitions per task? (L126)”
>
> **A6**: In BEHAVIOR, we want to address the fact that there is no single universal definition for these tasks. For example, one person might consider “putting away toys” to mean putting all the toys into a drawer, while another person might say it means placing plush toys on the shelf and plastic ones in a box. The initial conditions can also vary: even given a specific number of toys to start out with, one person might consider a good starting point to be strewn around the floor in a playroom, while someone else might consider that to mean the toys are in arbitrary locations around the house. All of these are valid and reflect the subjectivity and situationality of household activities, so rather than imposing one definition, we allow for multiple definitions per activity. We have developed an annotation tool (sec. A.3.3) to enable humans to generate definitions of each activity, and use our tool to collect two valid ones for each of the 100 activities. This is a different ethos than in other benchmarks, which often have a single definition per activity that may allow variation of position or object model, but not this more semantic variation.
>
> > **Q7**:“How can we compare different solutions in the efficiency metric? Comment on trade-offs, semantics of disarrangement...”
>
> **A7**: Great question! We have been thinking deeply about metrics and believe that our proposed metrics cover the most important axes of performance: energy, time, perturbations to the environment, etc.. However, the reviewer is right to point out that there are trade-offs: an agent could try to perform an activity faster but more “clumsy” or vice-versa. We believe this matches the situation in the real-world, where all these factors characterize different aspects of an agent. That is the reason we decided against combining all values in a single value for performance because this simplification would mask interesting insights about the agent’s performance.
> With respect to how to incorporate semantics into the metrics, this is another great question. This may require additional annotation, such as the material of an object. Concretely, for the example provided, adding the breakable logic state is in our agenda. The agent should learn to avoid breaking objects. But when several objects are breakable, such as a glass and a laptop, we may want to rank an agent that breaks only the glass as “less bad” than one that breaks the laptop. Our current set of metrics is comprehensive but there are other metrics that could be added in the future to account for other secondary objectives the agents should optimize for.

---

> ### Author Response · Authors · 2021-08-31
> **Response to Reviewer Ljtb (3/5)**
>
>
> > **Q3**: “how much different the low-level manipulations required are from eg object rearrangement”
>
> **A3**: This is a great question! There are two ways to look at it: from the side of actuation and from the side of the necessary changes in the environment the agent needs to cause to achieve the task.
> From the side of actuation, at the lowest level, there are just two types of physical manipulation a robot can perform: moving and/or applying forces [5,6]. Rearrangement tasks, and any other tasks in robotics benchmarks like RLBench, SoftGym or Metaworld [7,8,9], have to be solved with the same set of low-level manipulations: controlling robot motion and/or robot applied forces. So, all robot manipulation tasks are going to be solved either by moving or by applying forces on objects.
>
> When looking at what states of the objects the agent needs to learn to change, is where things are very different. All rearrangement tasks require the agent to reason how to change the pose of an object from A to B. In BEHAVIOR there are many additional changes in the environment that the agent needs to reason about, and learn to cause. For example, cleaning an object doesn’t mean moving it to another location, but rather by “removing all the stains” on the surface. Yes, the agent will use the same low-level manipulation (its motion and/or applied force) to wipe the surface by moving a wet wipe over it. The same for cooking an object: the agent/robot will use the same (only available) set of low-level manipulation capabilities: moving and applying forces on objects. But it needs to understand that it has to pick the food, place it over the stove and toggle the stove for the temperature of the food to change. Slicing food requires reasoning about how to use a tool (a knife or other cutting elements) even though the execution requires moving the knife and using it to apply force on the object to slice. These types of object changes the agent needs to learn to cause are much richer in BEHAVIOR and beyond the type of changes the agent needs to reason about in other benchmarks/types of tasks (see Figure A.2.b for a plot of the activities that require different types of state changes).
>
> > **C4**: “Comment on the “infeasibility of the tasks for current RL methods”, the alternative ”Use of demonstrations” to develop a solution, and the issue of the “large disparity in embodiment” between the embodiment used to collect VR data and the embodiment of a real robot.”
>
> **A4**: We agree that, given the results with state-of-the-art model-free RL algorithms, a promising avenue is to use human guidance in the form of the demonstrations. Our goal is to make use of the demonstrations to alleviate the long-horizon planning problem, while the low-level execution can vary from embodiment to embodiment.
> Since we only have a few (5) demonstrations per task, standard behavioral cloning does not provide robust long-horizon policies due to covariate shift. We are leveraging the human demonstrations to create high-level task-plans that can be used with motion primitives in iGibson to execute the plan with our agent.  This method uses a single human demonstration per task that is segmented into phases corresponding to the motion primitives (picking, placing, opening, navigating, etc) and executes the sequence of primitives with the AI agent. We hope to obtain preliminary results before the deadline.
> But the reviewer makes a great point: if the demonstrations provided by humans with the humanoid bimanual robot embodiment cannot be directly used for a robot, can we collect demonstrations directly with a robot embodiment? This is a good observation that we are pursuing in parallel in a different project line. In an independent submission to CoRL, we are exploring how humans can teleoperate robots to collect demonstrations with the same embodiment that the AI agent will control (the Fetch robot). This anonymous website provides more information and videos about this parallel effort (https://sites.google.com/view/il-for-mm/home). Our ultimate goal in that line of research is to be able to collect demonstrations for full BEHAVIOR tasks with this new interface to control mobile manipulators.
> Additionally, we believe there are other ways to perform morphological transference between different embodiments to be able to make some use of the VR demonstrations for imitation with another morphology [10, 11]. We will also explore these alternatives in the future.

---

> ### Author Response · Authors · 2021-08-31
> **Response to Reviewer Ljtb (2/5)**
>
> > **Q2**: “(2) on a higher level the paper lacks details what it means to "achieve" certain tasks” “How precise do I need to control the robot?” “Eg what is required to "load the dishwasher" or to "fill the stockings"?”
>
> **A2**: We apologize if this is not clear enough in the text and we have changed it to make it clear. In BEHAVIOR, achieving a task means fulfilling all the logical conditions included in the goal conditions in the activity definition. This definition has been given by a human annotator and they can be quite complex to reproduce in the main text but we have included a simplistic example in the Figure 1 (“Activity Goal Condition”) and a real example in Appendix, Listing 1 and 2. Reusing the simple example of Figure 1, in this “toyish” definition of “re-shelving library books”, achieving the task means that for all objects that are books, they are inside the shelf. Another task, “cleaning bathtub”, can be solved by navigating to the tub, grasping the brush (e.g. an oblong with bristles), navigating to the sink with brush in hand, “toggling on” the faucet by making contact between the hand and the toggle button, placing the brush in contact with a falling water particle so that it becomes “soaked”, navigating back to the tub, and moving the brush around so that its bristle side makes contact with at least 50% of the “stain” particles on the tub. This is on the simpler end of BEHAVIOR’s cleaning activities. This and two more examples, including “filling_Christmas_stockings”, have been added to sec. A.4. This link has a VR demonstration of “bottling_fruit”, to provide a visual example: https://streamable.com/1co5x0. Because the activities and their solution spaces are varied and vast, comprehensive analysis is difficult beyond what we have in secs. A.3 and A.6, but we hope these examples are helpful! More details about the implementation of the logic predicates (on, insideOf, nextTo, cooked, frozen, sliced…) are detailed in the iGibson 2.0 paper.
>
> In terms of how precise the control of the robot should be, our goal is to be as close as possible to the precision needed in the real world. Without the assistive grasping (see the reply before), the agent needs to interact in a very natural manner to solve the tasks. Opening a dishwasher requires the agent to move the hand and grasp the handle, then pull to open it. The robot embodiment and the shapes of the objects are realistic, so the precision of the motion is similar to the one in the real world. The outcome of the interaction should be as precise as needed to switch the corresponding logical condition from False to True. This will depend on the concrete condition. For example, for insideOf, the agent needs to move one object inside of the other; an object A is inside an object B if we can find a location inside object A to place a coordinate frame such that, when projecting a rays in opposing directions along the coordinate axes, the mesh of object B is hit for at least 2 out of 3 of the axes. For cooked to turn to True, the temperature of the cookable object should be brought over the cooking threshold (default 100°C). Additional concrete implementations of the predicates are included in the iGibson 2.0 manuscript in the Appendix.
>
> Beyond the concrete implementation, the main idea and objective is that, in BEHAVIOR, simulation and activity definitions are made to approximate the difficulty of the task in the real world. This is supported by the VR experience, where experienced users find the simulated activities are almost if not just as easy to complete as their real-world counterparts. All users find the initial and goal conditions to be realistic and plausible, and enforce a solution that would make sense in the real world.

---

> ### Author Response · Authors · 2021-08-31
> **Response to Reviewer Ljtb (1/5)**
>
> Thanks for the constructive feedback and we provide responses as below:
>
> > **Q1**: “What is the ‘realism of the required interactions for solving tasks’? ‘(1) on the very low-level: how realistically are interactions eg between the agent and objects simulated.’ Concretely, discuss assistive grasping.”
>
> **A1**: We agree that this is a critical element for a benchmark for embodied AI in simulation. Unlike other benchmarks such as Atari and DeepMind control suite, our goal is to get as realistic as possible in physical interaction. To that end, we base our work on a state-of-the-art physics engine, Bullet. Bullet is one of the most used simulators in robotics, and has demonstrated that it can be used to train policies that transfer to real-world with some sim2real adaptation [1,2,3,4]. All object-object and object-agent interactions are thus fully simulated, including kinematics, dynamics, friction, articulated objects, impacts, wheel navigation, etc.
>
> We understand the concern with respect to our assistive grasping (AG) mechanism: grasping is one of the hardest parts in many of these household activities and, in general, it is an active area in robotics research. We would like to clarify (and we have done in the main paper) that the assistive grasping is a mechanism available only to the humanoid bimanual robot so that the human experience in VR is as natural as possible. After several internal studies and multiple iterations, we observed that the strategies from humans in VR to grasp and perform the BEHAVIOR activities are more natural and closer to the real-world strategies (similar grasping modes, motion…) with a mechanism that makes grasping more robust (AG) than with a full simulation of the contact of the hand and fingers. This is due to the fact that the real human hand provides an extremely stable grasp, which is hard to obtain with a multi-finger hand in simulation. AG still forces the human agents to move the virtual hand to a “pregrasp” pose in a realistic manner, as it only triggers when the object is between the fingers and the palm (different from the common “sticky mitten” approximation in other simulators). AG is however not part of the Fetch robot embodiment, which grasps objects with a parallel jaw gripper (two fingers) with a full simulation of the physics. This grasping mode simulates more accurately the physical process involved when the real robot grasps an object, without any simplifications.

---

> ### Author Response · Authors · 2021-08-31
> **Follow-up response**
>
> Dear Reviewer:
>
> We apologize for the late post, we have been working hard to obtain empirical results that will clarify some of the reviewers' comments. These two additional replies relate to general questions about our submission:
>
> **On the use of VR data as imitation data to develop AI solutions:** As referenced in our earlier review, we have conducted an experiment to evaluate the utility of VR demonstrations to develop AI solutions. Using the logical predicate checking functions, we were able to extract kinematic and non-kinematic state changes for each human demonstration in order to segment the demonstrations into task plans. Segments correspond to some of the action primitives we have developed and provided as support for other researchers. For example, the “re-shelving library book” activity will have a task plan that looks like this: NavigateTo(book), Grasp(book), PlaceInside(book, shelf), etc. We then implemented partially simulated action primitives (see Appendix A.4) that allowed the agent to evaluate this task plan in the BEHAVIOR benchmark, and simulate the effect of the actions. Our agent successfully achieved 27 of the 238 task demos replayed, showing partial success in 85 of them. A partial success indicates that the agent successfully changed to True at least one of the predicates of the goal condition. Over all 238 demos, the agent switches to True 20% of the False predicates in the goal conditions.
>
> This is a promising first step to use human demonstrations as a source to learn high level plans. If the agent equipped with basic primitives is able to reproduce some of the success of the human agent, the VR demonstrations can be processed in this manner to learn to produce high-level plans. Future development in this line of research will need to improve the generation of task-plans from direct cloning them to learning to generate plans based on task conditions, and improve (possibly with data driven methods) the set of action primitives. Alternatively, these preliminary results should not discourage exploring other IL approaches that use the raw motion and interaction data in the VR demonstrations, without any intermediate action primitives; however, we believe this will be harder as the number of demonstrations per activity may not be not large enough to learn complex motion primitives.
>
> **On realistic interactions (using a realistic robot model)**: In addition, we would like to further illustrate the use of the simulated Fetch robot in BEHAVIOR. We have recorded a video of a teleoperator controlling the Fetch model. The user performs the "cleaning_table_after_clearing" BEHAVIOR activity to illustrate the use of a realistic robot model in BEHAVIOR tasks:
>
> https://streamable.com/pj7nu0
>
> While it is more difficult to grasp with a single fetch gripper, as can be seen in the video, the user is able to navigate through doors, pick up objects, trigger multiple non-kinematic state changes (toggling on the faucet and soaking the cloth). In this video, grasping is fully simulated with rigid body contacts between the two fingers and the objects. This embodiment is fully functional and available for the benchmark, modeled as close as possible to the real robot to provide a more realistic interaction and strategies that are close to the ones required by real robots.
>
> We hope this further clarifies some of the questions and comments by the reviewers.
>
> Best regards,
>
> Authors of CoRL 2021 Conference Paper87

---

### Official Review · Reviewer_NSMD · 2021-07-24

**Originality:** Very Good
**Technical Quality:** Good
**Clarity Of Presentation:** Fair
**Impact:** 4

**Recommendation:**

Weak Accept: I recommend accepting the paper, but will not argue for my recommendation if the majority of other reviewers have a different opinion.

**Summary:**

This work partially describes a novel benchmark of human VR demonstrations for everyday household activities.  The activities chosen are for actual common tasks (based on empirical frequency rankings).  Tasks are embodied within iGibson and coded for with a new logical description language. The work is situated alongside a quickly growing collection of embodied AI benchmarks and several of the new tasks are used for training baseline RL systems.

**Issues:**

I would appreciate addressing any of the weaknesses above and incorporating additional technical details in the primary paper, perhaps by removing unnecessary details about iGibson, selling of the dataset, and streamlining the modeling discussion.

**Reviewer Expertise:**

Very good: Comprehensive knowledge of the area

**Strengths And Weaknesses:**

**Strengths**:
- The demonstration dataset is based on real action sequences collected at scale from humans.
- The data includes long horizon tasks with complex dynamics and are highly diverse within the home setting.

**Weaknesses**:
The current report spends more energy selling the concept of the dataset than providing useful or necessary details.  There are many places where a claim is made, decision taken, or statistic shown but then no follow-up throughout the paper explains it.  I'll note a few such cases here:
1. "Activity Length: 20,000" -- What activity is 20,000 steps long? and this is simply an error or something meaningful? What is the distribution or std dev?
2. "unlimited human-defined versions" and "infinite unique instantiations" -- is this simply saying that surfaces are technically R^2? This is therefore true of several other environments.
3. Section 5 outlines the primitives, this cross product is probably actually the accurate measure of the size and diversity of the dataset?
4. Why evaluate only the 12 most simple activities? and if compute was the bottleneck, why wasn't a simple imitation learning policy used first on a larger set of activities instead of RL?
5. If the primary difficulty is due to physics, it seems like we would want to specifically train a physics model (supervised) and incorporate it here? Once we've done this, if many of the simple tasks are now able to very quickly perform at ceiling, a more complete evaluation could be run efficiently over the full dataset?

Minor clarifications:
1. In the current writing it is often difficult to determine which platform and action space is in use.  Minimally, am I correct in understanding that no results use the simulated Fetch platform?
2. The continuous action space does not include fingers? So grasps are still effectively binary motion primitives here?  Can you therefore elaborate on what aspects of the physics prove most difficult?

**Summary Of Recommendation:**

I think the dataset collected has very serious potential for the community and I very much appreciate the effort and energies that have gone into its construction.  I do not however feel that the current manuscript does the work service.  I learned and enjoyed reading more the appendix of this paper than the actual paper.  This may be partially the fault of the 8-page limit, but I think the authors are attempting to do too many things in one paper: 1. Introduce new abilities of iGibson, 2. Introduce BDDL, 3. Sell a dataset as a task, 4. Define naive models, 5. Analyze said models.  4 and 5 felt particularly weak since they have such low coverage and physics is the sole difficulty.  The effect is that too many details (e.g. about values in Table 1) are never properly discussed.


Update: Thank you for a genuine effort to respond to all points and address the questions.  The responses definitely helped me better understand aspects of the data (and action space) that were unclear to me in the original reading.   My comment about selling may be partially due to the fact that I am already more than sold the area of research and the potential utility of the data, so the high-level discussion felt like it used up space that would be better used to expand on all the details in the replies here.  A future ArXiv version could include a more complete version of the paper.  A minor note, distributions for task types and overall for the data would be appreciated.  While the explanation of the very long trajectory makes sense, I think it is still misleading in the table.  Even just adding a column with the average would be helpful.

---

> ### Author Response · Authors · 2021-08-31
> **Response to Reviewer NSMD (4/4)**
>
> > **C8**: The continuous action space does not include fingers? So grasps are still effectively binary motion primitives here? Can you therefore elaborate on what aspects of the physics prove most difficult?
>
> **A8**: The continuous space does include fingers but not independently. They are actuated all together in a synchronized closing motion controlled by a one degree of freedom actuation per hand. In VR, this DoF is mapped to the continuous trigger in the controller. Similarly, the Fetch robot has two fingers controlled by a one degree of freedom for both to close/open as it is the case on the real-robot. What is binary is the assistive grasping that activates after a certain threshold of the one degree of freedom of the closing action is surpassed, and if certain conditions are met (e.g. there is an object between the fingers and the palm).
>
> With respect to “what aspects of physics prove most difficult” for us (VR users), grasping without assistive grasping was extremely challenging due to the multi-contact system created between the hand and the objects. Assistive grasping definitively improved usability and removed physics artifacts such as penetration of the fingers inside object surfaces. This effect (penetrations) is also difficult to eliminate in the physics simulator. While other works have increased the physics timestep to accelerate simulation, we found that doing that creates multiple penetrations between the agent and the objects or between pairs of objects. We decreased the simulation timestep to obtain realistic agent-object and object-object interactions.
>
> With respect to “what aspects of physics prove most difficult” for the agent, controlling physical interactions (contact, forces) to obtain the desired object motion is the greatest challenge. While large, fixed articulated objects are easier to interact with (their kinematic constraints help the agent guide the interaction to the allowed degree of freedom to actuate them), smaller objects are hard to push to reach other objects behind them. Moreover, while assistive grasping facilitates the grasp and therefore creating the desired motion for small objects, placing these objects in the right configuration, e.g. straight up, or between other objects without knocking them, is not easy for the agents. The correct use of tools (knives, cleaning tools) is also hard, e.g., it is not always easy to fixate one object to slice it by applying enough force with the sharp side of the knife on it. Sometimes even the physical execution of a complex navigation trajectory with our full physics simulation reveals more challenges than if physics is deactivated because the agent touches objects that may fall (e.g. a lamp, objects on a table) creating undesired disarrangement. These are some examples we observed of the difficulties encountered when fully simulating the physical interactions.
>
> ### References
> - [1] Haarnoja, T., Zhou, A., Abbeel, P., & Levine, S. (2018, July). Soft actor-critic: Off-policy maximum entropy deep reinforcement learning with a stochastic actor. In International conference on machine learning (pp. 1861-1870). PMLR.
> - [2] Haarnoja, T., Zhou, A., Hartikainen, K., Tucker, G., Ha, S., Tan, J., ... & Levine, S. (2018). Soft actor-critic algorithms and applications. arXiv preprint arXiv:1812.05905.
> - [3] Wahid, A., Stone, A., Chen, K., Ichter, B., & Toshev, A. (2020). Learning object-conditioned exploration using distributed soft actor critic. Conference on Robot Learning.

---

> ### Author Response · Authors · 2021-08-31
> **Response to Reviewer NSMD (3/4)**
>
> > **Q5**: “Why evaluate only the 12 most simple activities? and if compute was the bottleneck, why wasn't a simple imitation learning policy used first on a larger set of activities instead of RL?”
>
> **A5**: Yes, computation was a bottleneck. We wanted to provide results with a state-of-the-art method that has achieved good results in other embodied AI / robotics tasks, and SAC [1, 2, 3] is currently one of the most used algorithms by the community. We believe these results inform about what could be expected by applying state-of-the-art model-free RL on BEHAVIOR tasks, even the most simple ones. Additionally, our ablations provide insights on what makes BEHAVIOR activities so hard for a method that has shown such impressive results in other tasks.
> The reviewer is right in that an alternative path is to use the collected human demos for imitation learning, and we are actively exploring this path. However, since we only have a few (5) demonstrations per task, standard behavioral cloning fails to provide robust long-horizon policies that do not diverge due to covariate shift. The alternative we are exploring is to leverage the human demonstrations at a “higher level” to obtain a task-plan, while relying on primitives at the lower level for the execution of the plan. More concretely, we are working on a solution that uses one human demonstration per task, segments the demo into phases that match the primitives we provide (picking, placing, opening, navigating, etc) and executes the sequence of primitives with the AI agent. We are currently working on this solution and hope to obtain preliminary results that we can post here before the deadline.
>
> > **Q6**: “If the primary difficulty is due to physics, it seems like we would want to specifically train a physics model (supervised) and incorporate it here?”
>
> **A6**: Our experiments indicate that the primary difficulty is controlling the environment through physical interactions with the embodiment to achieve the physical task. A model-based approach may help here. However, it is unclear if the complex physics during mobile manipulation could be learned and used for planning effectively, and then controlled to achieve the desired outcome. Thank you for the suggestion. This is definitively a good strategy that we will explore in the future, but due to the complexity of learning powerful physics models for realistic interactions, we consider this may go beyond the scope of the current paper.
>
> > **C7**: In the current writing it is often difficult to determine which platform and action space is in use. Minimally, am I correct in understanding that no results use the simulated Fetch platform?
>
> **A7**: Yes, this is correct. For consistency and comparability, we have used the bimanual humanoid robot in all our experiments. The humanoid bimanual robot, with two manipulators resembling human hands, is also more natural for VR participants to use. However, the Fetch robot is fully functional and could be alternatively used. We have modified the text to clarify the embodiment and action space used in each experiment.

---

> ### Author Response · Authors · 2021-08-31
> **Response to Reviewer NSMD (2/4)**
>
> > **Q3**: “‘unlimited human-defined versions’ and ‘infinite unique instantiations’ -- is this simply saying that surfaces are technically R^2? This is therefore true of several other environments.”
>
> **A3**: Thanks for the question, there are two elements here to clarify on this point:
>
> **Unlimited human-defined versions**: In BEHAVIOR we want to acknowledge that there is no unique definition for each task. For example, one person might consider “putting_away_toys” to mean putting all the toys into a drawer, while another person might say it means placing plush toys on the shelf and plastic ones in a box. The initial conditions can also vary: even given a specific number of toys to start out with, one person might consider a good starting point to be strewn around the floor in a playroom, while someone else might consider that to mean the toys are in arbitrary locations around the house. This is different from the other benchmarks that have only a single valid definition per activity. To account for these potentially unlimited human-defined versions, in BEHAVIOR we have developed an annotation tool (Appendix A3.3) that enables humans to easily generate definitions per activity. Using our tool we have generated two valid definitions for each of the 100 activities. The tool, which we will make available online after the review to keep collecting definitions, would allow us to collect “unlimited” (bounded by the required time) definitions per activity.
>
> **Infinite unique instantiations**: An instantiation is a concrete set of state values for each object in a scene that fulfills the semantically defined activity initial state, for example, by sampling locations for “N1 toys on the floor of the children’s room”. The reviewer is correct in saying that other benchmarks can create potentially infinite instances by sampling in predefined areas. What sets BEHAVIOR apart is that our definitions are semantic, not geometric, and valid for multiple scenes. The same BDDL definition can yield infinite instances in any scene that has the requested furniture, room, and space for objects without the time-consuming labor of manually defining sampling areas for each scene, even if the geometries of these scene layouts vary significantly. Additionally, the semantics combined with the taxonomically organized objects make the definitions more general. For example, consider an initial state defined as “inside(fruit1, cabinet1)”. Several different fruits can be sampled, and in different cabinets each time, for any scene with cabinets. We believe that this is a valuable and unique feature of BEHAVIOR that distinguishes it from existing benchmarks and environments. This is reflected in Table A.1, row “infinite scene-agnostic instantiation”.
>
> We have modified the text in the main paper to make this point clearer.
>
> > **Q4**: “Section 5 outlines the primitives, this cross product is probably actually the accurate measure of the size and diversity of the dataset?”
>
> **A4**: We are not completely sure of what is referred to here with “primitives” of Section 5 but we assume that they are the functional requirements for a simulator to implement BEHAVIOR. It is not possible to “cross product” some of these requirements as they are not countable, e.g. to provide an object-centric representation with additional properties beyond kinematics, or to be able to sample semantically defined activities. For the ones that can be counted (e.g., object models, scenes, agents, activities, activity definitions, object states…) we could provide a cross-product but we are not sure if the number would be informative or help to compare “apples to apples” to other benchmarks. As mentioned above, thanks to the BDDL-semantic language used in the definitions of BEHAVIOR and fully instantiated in iGibson 2.0, one single activity definition represents a diversity that is hard to quantify and compare to other benchmarks that do not enable sampling or where the sampling is restricted to manually annotated geometric areas. In any case, this is a very valuable suggestion to attempt to quantify diversity.

---

> ### Author Response · Authors · 2021-08-31
> **Response to Reviewer NSMD (1/4)**
>
> Thanks for the constructive feedback and we provide responses as below:
>
> > **C1**: “Clarifying some details in the main part of the paper”
>
> **A1**: We agree with the reviewer that it is hard to summarize this work in 8 pages. We tried our best but we are grateful for all suggestions to make the paper clearer. The suggestions from the reviewer to trim for the main paper are:
> removing unnecessary details about iGibson
> selling of the dataset
> streamlining the modeling discussion
>
> With respect to iGibson, we have included only the minimum necessary information (one paragraph about the agents, sensing, action primitives) to understand the experiments without having to jump to the appendix. We are afraid that removing that paragraph may break readability, but we have tried to reduce this section in the new version of the paper.
>
> With respect to “selling the dataset”, we believe that to value the work, the readers need to first understand the need. As other reviewers mentioned, the landscape of benchmarks and simulators is rich: we try to justify the need for BEHAVIOR and for the design decisions we made, instead of just describing what we did.
>
> We can’t figure out what modeling discussion is referred to here (objects, activities, RL models?), but we are very much open to suggestions to streamline it and improve it. Could the reviewer provide some indications of which parts are not necessary to understand the rest of the paper?
>
> In the following, some clarifications to the concrete questions:
>
> > **Q2**: “What activity is 20,000 steps long?”
>
> **A2**: We count as steps here the time instants where the agent needs to make a decision based on observations. Using the MDP formalism, an activity is N steps long if this is the length of the sequence of actions a_i that the agent needs to take to transition from the initial state s_0 to a valid goal state. The numbers we use here are for human demonstrators, which we assume to be close to an optimal solution. In the case of 20000, the activity is “packing_picnics”, where one of the human operators required 11 minutes to finish the activity (20000 steps, controlling the agent at 30fps, 1/30 s between steps). While arguably this is not the most optimal solution, it is simply a complex task where many objects have to be found, prepared, and packed together.
>
> This value is a max value from all demonstrations from humans. The minimum is 300 steps (10 seconds). The complete durations for all activities are shown in Figure A.12.a. The mean value is also shown there, 5580 steps (standard deviation = 3645).
>
> We think steps are a useful unit to compare different benchmarks from the point of view of the length of the task. In some benchmarks the agents need to control low level actions (high frequency), in others, they can use high-level actions (action primitives) that execute for a longer time. Using wall-clock time to compare task horizons would be misleading about the actual planning complexity and length of the decision making problem. With steps, we aim to communicate how many decisions/actions need the agent to take to reach a goal.

---

> ### Author Response · Authors · 2021-08-31
> **Follow-up response**
>
> Dear Reviewer:
>
> We apologize for the late post, we have been working hard to obtain empirical results that will clarify some of the reviewers' comments. These two additional replies relate to general questions about our submission:
>
> **On the use of VR data as imitation data to develop AI solutions:** As referenced in our earlier review, we have conducted an experiment to evaluate the utility of VR demonstrations to develop AI solutions. Using the logical predicate checking functions, we were able to extract kinematic and non-kinematic state changes for each human demonstration in order to segment the demonstrations into task plans. Segments correspond to some of the action primitives we have developed and provided as support for other researchers. For example, the “re-shelving library book” activity will have a task plan that looks like this: NavigateTo(book), Grasp(book), PlaceInside(book, shelf), etc. We then implemented partially simulated action primitives (see Appendix A.4) that allowed the agent to evaluate this task plan in the BEHAVIOR benchmark, and simulate the effect of the actions. Our agent successfully achieved 27 of the 238 task demos replayed, showing partial success in 85 of them. A partial success indicates that the agent successfully changed to True at least one of the predicates of the goal condition. Over all 238 demos, the agent switches to True 20% of the False predicates in the goal conditions.
>
> This is a promising first step to use human demonstrations as a source to learn high level plans. If the agent equipped with basic primitives is able to reproduce some of the success of the human agent, the VR demonstrations can be processed in this manner to learn to produce high-level plans. Future development in this line of research will need to improve the generation of task-plans from direct cloning them to learning to generate plans based on task conditions, and improve (possibly with data driven methods) the set of action primitives. Alternatively, these preliminary results should not discourage exploring other IL approaches that use the raw motion and interaction data in the VR demonstrations, without any intermediate action primitives; however, we believe this will be harder as the number of demonstrations per activity may not be not large enough to learn complex motion primitives.
>
> **On realistic interactions (using a realistic robot model)**: In addition, we would like to further illustrate the use of the simulated Fetch robot in BEHAVIOR. We have recorded a video of a teleoperator controlling the Fetch model. The user performs the "cleaning_table_after_clearing" BEHAVIOR activity to illustrate the use of a realistic robot model in BEHAVIOR tasks:
>
> https://streamable.com/pj7nu0
>
> While it is more difficult to grasp with a single fetch gripper, as can be seen in the video, the user is able to navigate through doors, pick up objects, trigger multiple non-kinematic state changes (toggling on the faucet and soaking the cloth). In this video, grasping is fully simulated with rigid body contacts between the two fingers and the objects. This embodiment is fully functional and available for the benchmark, modeled as close as possible to the real robot to provide a more realistic interaction and strategies that are close to the ones required by real robots.
>
> We hope this further clarifies some of the questions and comments by the reviewers.
>
> Best regards,
>
> Authors of CoRL 2021 Conference Paper87

---

### Meta-Review · Area_Chair_joFL · 2021-08-13

**Recommendation:** Accept (Poster)
**Confidence:** 4

**Metareview:**

**Update after rebuttal**
I thank the authors for their thorough responses, my concerns/questions have been addressed (and the reviewers concerns as well). I recommend accept.

**Initial meta-review**
**Summary**:
This work proposes a benchmark for every day household activities/tasks, as well as presents a dataset of human (VR) demonstrations that show how to perform these tasks. The proposed framework is independent of a specific simulator, and in this work has been instantiated via iGibson. The authors identify the following contributions
* a high-level PDDL-style language which enables the generation of arbitrary many instances of a task
* Identification of environment-agnostic functional requirements, such that the proposed framework can be instantiated in any environment/simulator that fulfills these requirements. A specific example is given through instantiation in iGibson
* Developing a comprehensive set of metrics for consistent evaluation across various tasks. Specifically a set of metrics relative to human demonstrations (collected in VR) are proposed. The human demonstrations can also be utilized for evaluating imitation learning.

**Strengths**:

* This work addresses an important and very timely problem of benchmarking and evaluating the growing literature on embodied AI research. All reviewers agree that this work could have significant impact.
* Reviewers appreciate the complexity of the proposed proposed benchmark household activities and the engineering effort that must have gone into setting this up as well as combining it with human demonstrations. Furthermore it is appreciated that the proposed framework is inherently simulation independent
* The reviewers mostly agree that this manuscript is well written and easy to follow

**Weaknesses**:

* One of the claimed contributions are the PDDL-style language for describing tasks. However, this manuscript does not delineate how it differs from other PDDLs. While the authors cite one such work, the authors merely mention that the one proposed in this work is different from the cited one. As one of your main contributions this part of the related work needs to be discussed more thoroughly. A lot of work exists in this area and at least some of it should be referenced. A good entry point may be “A Review and Comparison of Ontology-based Approaches to Robot Autonomy “ https://upcommons.upc.edu/bitstream/handle/2117/179204/2019_KER_A_Review_and_Comparison_of_Ontology-based_Approaches_to_Robot_Autonomy_OlicaresEtAl.pdf
* While the reviewers agree that the manuscript is easy to follow, they have  also identified several places that lack important details. The authors should address these questions/lack of details mentioned by the reviewers
* A big question is the question of realism. While realism is addressed in the manuscript, it only addresses high-level realism (at the level of task description). The paper lacks a clear description of what level of realism is present in the execution of each of the subtasks. For instance, what does grasping look like? Especially for a conference like CoRL it would be important to discuss how well results obtained simulated benchmark would transfer to the real world.

---

> ### Author Response · Authors · 2021-08-31
> **Response to Area Chair joFL (3/3)**
>
> Regarding the use of PDDL in ALFRED: The main common ground is that the languages have a very similar structure of initial and goal conditions. There are also some key distinctions: like other uses of PDDL, ALFRED’s logical operators are a subset of BDDL’s, as detailed above for PDDLs in general. Another difference is that ALFRED’s use for PDDL is to annotate action plans from demonstrations and use planners much like in other applications of PDDL, whereas BDDL does not specify plans and describes only world states, i.e., there are no action symbols. This speaks to a fundamental difference in the purpose of having a domain-specific language, as we see BDDL as a usable, flexible, and extensible way to define naturalistic activities beyond its uses in planning.
>
> ### References
> - [1] Olivares-Alarcos, A., Beßler, D., Khamis, A., Goncalves, P., Habib, M. K., Bermejo-Alonso, J., ... & Li, H. (2019). A review and comparison of ontology-based approaches to robot autonomy. The Knowledge Engineering Review, 34.
> - [2] Gil, Y. (2005). Description logics and planning. AI magazine, 26(2), 73-73.
> - [3] Iscen, A., Caluwaerts, K., Tan, J., Zhang, T., Coumans, E., Sindhwani, V., & Vanhoucke, V. (2018, October). Policies modulating trajectory generators. In Conference on Robot Learning (pp. 916-926). PMLR.
> - [4] Peng, X. B., Coumans, E., Zhang, T., Lee, T. W., Tan, J., & Levine, S. (2020). Learning agile robotic locomotion skills by imitating animals. In Robotics Science and Systems 2020.
> - [5] Yu, W., Tan, J., Bai, Y., Coumans, E., & Ha, S. (2020). Learning fast adaptation with meta strategy optimization. IEEE Robotics and Automation Letters, 5(2), 2950-2957.
> - [6] Seita, D., Florence, P., Tompson, J., Coumans, E., Sindhwani, V., Goldberg, K., & Zeng, A. (2021). Learning to Rearrange Deformable Cables, Fabrics, and Bags with Goal-Conditioned Transporter Networks. IEEE International Conference on Robotics and Automation (ICRA).

---

> ### Author Response · Authors · 2021-08-31
> **Response to Area Chair joFL (2/3)**
>
> > **C3**: Similarities and differences between BDDL and PDDL
>
> **A3**: This is a great point. As pointed out, BDDL is indeed similar to PDDL as both are logic languages, and BDDL takes inspiration from PDDL committing to a similar syntax to facilitate its use and adoption by researchers familiar with PDDL. We thank the meta-reviewer for the helpful entrypoint reference Olivares-Alarcos et al. [1], and would like to point to the additional reference of Gil [2], which discusses knowledge in description logics like PDDL. Gil’s discussion of object and goal representation highlights the structural similarity between BDDL and various PDDLs, its discussion of action and plan knowledge highlights the difference in purpose between them, and Olivares-Alarcos et al.’s discussion of the mathematical claims of planning languages sets up the difference in logical operators between them. These differences are to allow intuitive and flexible human-generated definitions in BDDL without considering planning. We elaborate on motivations and details of these below.
>
> BDDL’s syntax, domain structure, and problem structure are similar to standard PDDL. Various description logics use objects in a type hierarchy [2], like BDDL’s domain objects; and state descriptions as ground predicates [2], much like BDDL’s ground initial condition representing a single symbolic state. Gil also describes how some goal representations in some PDDLs are predicates satisfied by unification, whereas others are taxonomic and possibly adaptive. BDDL goals are compositional expressions, satisfied by unification. As in all of the PDDLs described in Gil, BDDL goals are completely declarative.
>
> A key difference between BDDL and many PDDLs is their purpose. PDDLs generally aim to enable symbolic planning [1, 2]. They, therefore, have rules to define both state and action symbols, and all necessary elements (e.g. transition functions between world states) that allow planning algorithms to find action sequences from initial to goal state [2]. Meanwhile, BDDL aims to characterize the state of the environment in a semantic form. This facilitates the generation of general definitions for the 100 activities as initial and goal symbolic states. Therefore, BDDL, together with the implementations of condition checking and sampling, maps between the continuous physical state and the symbolic state, but it does not have action symbols, so that solutions to such complex activities are not limited to actions the authors could come up with. All transitions/forward models are the simulated physical processes that an agent controls via physical interactions. This motivation is validated by the fact that the VR demonstrations themselves show many creative ways of transporting objects, dusting, arranging, etc. that would be difficult to predefine exhaustively. The only actions in BEHAVIOR are robot control commands. Therefore, unlike PDDL, BDDL should not be called a “planning language”.
>
> Additionally, in terms of syntax, PDDLs’ logic operators obey the axioms of first-order logic [1]. Differently, BDDL’s quantifiers are a superset of standard PDDL’s, with the additional quantifiers, ForPairs, ForN, and ForNPairs, only obeying first-order logic for certain sets of objects. These three additional quantifiers are designed to simplify the process of defining conditions facilitating the crowdsourcing work for activity definitions by non-PDDL experts. For example, *ForPairs: apple, bag. inside(apple, bag)* says that some one-to-one pairing of apples and bags must be made where each apple is in its corresponding bag. This is more general than saying e.g. “inside(apple1, bag1) AND inside(apple2, bag2)....” and helps annotators avoid such a brittle scenario. More details on the definitions and utility of these are in sec. A.3.2.
>
> Overall, PDDL and BDDL have many similarities, but they have different purposes. Since BDDL does not obey completely the formalism of first-order logic and prioritizes instead usability for crowdsourcing, and BDDL does not include action symbols to enable planning but only state symbols, we do not consider it a version of PDDL but rather an alternative language.
>
> For a specific comparison to ALFRED as requested by BvZu, please see the below paragraph which also appears in our response to them. We have added explanations and references to these papers in A.3.2.

---

> ### Author Response · Authors · 2021-08-31
> **Response to Area Chair joFL (1/3)**
>
> We thank all of the reviewers and the meta-reviewer for their detailed and insightful comments. We are glad to see that the reviewers and AC consider BEHAVIOR to “address an important and very timely problem of benchmarking”, value the “engineering effort” and that the work is “simulation independent”, and find the manuscript “well written and easy to follow”. The concerns and comments are very insightful: we have made revisions to the paper and supplementary material (highlighted in red, and pointed out in our responses), and replied to each reviewer’s comments. Please note that some red highlighted sections may look quite long, but some of these are revisions of existing text with small changes throughout, not necessarily entirely new/rewritten sections. We are also working hard to include as many of the suggested additional experiments as possible in the two-week rebuttal period.
>
> In the following, we will reply to the reviewers one by one. Please let us know if we misinterpreted any of the suggestions/questions and we will further clarify. Thanks!
>
> > **C1**: Address lack of details and questions raised by the reviewers
>
> **A1**: We included replies to the comments of the reviewers including the mentioned lack of details and made requisite changes to the paper.
>
> > **C2**: Discussion on realism, especially at the lowest level, e.g., grasping
>
> **A2**: We acknowledge the importance of realism at the lowest level to understand the relevance of BEHAVIOR for robotics and for CoRL. We have included an extensive discussion on realism in our reply to Ljtb that we reproduce here:
>
> We agree that this is a critical element for a benchmark for embodied AI in simulation. Unlike other benchmarks such as Atari and DeepMind control suite, our goal is to get as realistic as possible in physical interaction. To that end, we base our work on a state-of-the-art physics engine Bullet. Bullet is one of the most used simulators in robotics, and has demonstrated that it can be used to train policies that transfer to real world with some sim2real adaptation [3,4,5,6]. All object-object and object-agent interactions are thus fully simulated, including kinematics, dynamics, friction, articulated objects, impacts, wheel navigation, etc.
>
> We understand the concern with respect to our assistive grasping (AG) mechanism: grasping is one of the hardest parts in many of these household activities and, in general, it is an active area in robotics research. We would like to clarify (and we have done in the main paper) that assistive grasping is a mechanism available only to the humanoid bimanual robot so that the human experience in VR is as natural as possible. After several internal studies and multiple iterations, we observed that the strategies from humans in VR to grasp and perform the BEHAVIOR activities are more natural and closer to the real-world strategies (similar grasping modes, motion…) with a mechanism that makes grasping more robust (AG) than with a full simulation of the contact of the hand and fingers. This is due to the fact that the real human hand provides an extremely stable grasp, which is hard to obtain with a multi-finger hand in simulation. AG still forces the human agents to move and “pregrasp” in a realistic manner, as it only triggers when the object is between the fingers and the palm (different to the common “sticky mitten” approximation in other simulators). AG is however not part of the Fetch robot embodiment, which grasps objects with a parallel jaw gripper (two fingers) with a full simulation of the physics. This grasping mode simulates more accurately the physical process involved when the real robot grasps an object, without any simplifications.

---

> ### Author Response · Authors · 2021-08-31
> **Follow-up response to Area Chair**
>
> Dear Area Chair,
>
> We apologize for the late post, we have been working hard to obtain empirical results that will clarify some of the reviewers' comments. These two additional replies relate to general questions about our submission:
>
> **On the use of VR data as imitation data to develop AI solutions:** As referenced in our earlier review, we have conducted an experiment to evaluate the utility of VR demonstrations to develop AI solutions. Using the logical predicate checking functions, we were able to extract kinematic and non-kinematic state changes for each human demonstration in order to segment the demonstrations into task plans. Segments correspond to some of the action primitives we have developed and provided as support for other researchers. For example, the “re-shelving library book” activity will have a task plan that looks like this: NavigateTo(book), Grasp(book), PlaceInside(book, shelf), etc. We then implemented partially simulated action primitives (see Appendix A.4) that allowed the agent to evaluate this task plan in the BEHAVIOR benchmark, and simulate the effect of the actions. Our agent successfully achieved 27 of the 238 task demos replayed, showing partial success in 85 of them. A partial success indicates that the agent successfully changed to True at least one of the predicates of the goal condition. Over all 238 demos, the agent switches to True 20% of the False predicates in the goal conditions.
>
> This is a promising first step to use human demonstrations as a source to learn high level plans. If the agent equipped with basic primitives is able to reproduce some of the success of the human agent, the VR demonstrations can be processed in this manner to learn to produce high-level plans. Future development in this line of research will need to improve the generation of task-plans from direct cloning them to learning to generate plans based on task conditions, and improve (possibly with data driven methods) the set of action primitives. Alternatively, these preliminary results should not discourage exploring other IL approaches that use the raw motion and interaction data in the VR demonstrations, without any intermediate action primitives; however, we believe this will be harder as the number of demonstrations per activity may not be not large enough to learn complex motion primitives.
>
> **On realistic interactions (using a realistic robot model)**: In addition, we would like to further illustrate the use of the simulated Fetch robot in BEHAVIOR. We have recorded a video of a teleoperator controlling the Fetch model. The user performs the "cleaning_table_after_clearing" BEHAVIOR activity to illustrate the use of a realistic robot model in BEHAVIOR tasks:
>
> https://streamable.com/pj7nu0
>
> While it is more difficult to grasp with a single fetch gripper, as can be seen in the video, the user is able to navigate through doors, pick up objects, trigger multiple non-kinematic state changes (toggling on the faucet and soaking the cloth). In this video, grasping is fully simulated with rigid body contacts between the two fingers and the objects. This embodiment is fully functional and available for the benchmark, modeled as close as possible to the real robot to provide a more realistic interaction and strategies that are close to the ones required by real robots.
>
> We hope this further clarifies some of the questions and comments by the reviewers.
>
> Best regards,
>
> Authors of CoRL 2021 Conference Paper87

---

### Decision · Program_Chairs · 2021-09-13

**Decision:**

Accept (Poster)

**Comment:**

**Update after rebuttal**
I thank the authors for their thorough responses, my concerns/questions have been addressed (and the reviewers concerns as well). I recommend accept.

**Initial meta-review**
**Summary**:
This work proposes a benchmark for every day household activities/tasks, as well as presents a dataset of human (VR) demonstrations that show how to perform these tasks. The proposed framework is independent of a specific simulator, and in this work has been instantiated via iGibson. The authors identify the following contributions
* a high-level PDDL-style language which enables the generation of arbitrary many instances of a task
* Identification of environment-agnostic functional requirements, such that the proposed framework can be instantiated in any environment/simulator that fulfills these requirements. A specific example is given through instantiation in iGibson
* Developing a comprehensive set of metrics for consistent evaluation across various tasks. Specifically a set of metrics relative to human demonstrations (collected in VR) are proposed. The human demonstrations can also be utilized for evaluating imitation learning.

**Strengths**:

* This work addresses an important and very timely problem of benchmarking and evaluating the growing literature on embodied AI research. All reviewers agree that this work could have significant impact.
* Reviewers appreciate the complexity of the proposed proposed benchmark household activities and the engineering effort that must have gone into setting this up as well as combining it with human demonstrations. Furthermore it is appreciated that the proposed framework is inherently simulation independent
* The reviewers mostly agree that this manuscript is well written and easy to follow

**Weaknesses**:

* One of the claimed contributions are the PDDL-style language for describing tasks. However, this manuscript does not delineate how it differs from other PDDLs. While the authors cite one such work, the authors merely mention that the one proposed in this work is different from the cited one. As one of your main contributions this part of the related work needs to be discussed more thoroughly. A lot of work exists in this area and at least some of it should be referenced. A good entry point may be “A Review and Comparison of Ontology-based Approaches to Robot Autonomy “ https://upcommons.upc.edu/bitstream/handle/2117/179204/2019_KER_A_Review_and_Comparison_of_Ontology-based_Approaches_to_Robot_Autonomy_OlicaresEtAl.pdf
* While the reviewers agree that the manuscript is easy to follow, they have  also identified several places that lack important details. The authors should address these questions/lack of details mentioned by the reviewers
* A big question is the question of realism. While realism is addressed in the manuscript, it only addresses high-level realism (at the level of task description). The paper lacks a clear description of what level of realism is present in the execution of each of the subtasks. For instance, what does grasping look like? Especially for a conference like CoRL it would be important to discuss how well results obtained simulated benchmark would transfer to the real world.